# LRRTM2 controls presynapse nano-organization and AMPA receptor sub-positioning through Neurexin-binding interface

Konstantina Liouta [1,2], Malgorzata Lubas[1,2,6], Vasika Venugopal [1,2,6], Julia Chabbert[1,2], Caroline Jeannière[1,2], Candice Diaz[1,2], Matthieu Munier[1,2], Béatrice Tessier [1,2], Stéphane Claverol [3], Alexandre Favereaux[1,2], Matthieu Sainlos[1,2], Joris de Wit [4,5], Mathieu Letellier [1,2], Olivier Thoumine [1,2] & Ingrid Chamma [1,2] ✉

Synapses are organized into nanocolumns that control synaptic transmission efficacy through precise alignment of postsynaptic neurotransmitter receptors and presynaptic release sites. Recent evidence show that Leucine-Rich Repeat Transmembrane protein LRRTM2, highly enriched and confined at synapses, interacts with Neurexins through its C-terminal cap, but the role of this binding interface has not been explored in synapse formation and function. Here, we develop a conditional knock-out mouse model (cKO) to address the molecular mechanisms of LRRTM2 regulation, and its role in synapse organization and function. We show that LRRTM2 cKO specifically impairs excitatory synapse formation and function in mice. Surface expression, synaptic clustering, and membrane dynamics of LRRTM2 are tightly controlled by selective motifs in the C-terminal domain. Conversely, the N-terminal domain controls presynapse nano-organization and postsynapse AMPAR sub-positioning and stabilization through the recently identified Neurexin-binding interface. Thus, we identify LRRTM2 as a central organizer of pre- and post-excitatory synapse nanostructure through interaction with presynaptic Neurexins.

Information processing in the brain critically relies on proper neuronal connectivity. Synapses are highly specialized macromolecular platforms containing hundreds of distinct proteins that co-organize across neuronal development and circuit maturation, to provide stable yet plastic sites of information processing[1,2]. Advances in super-resolution microscopy and the development of compatible probes have significantly contributed to dissecting synapse organization and function[3–5]. Increasing evidence supports that the release machinery is aligned with neurotransmitter receptors, forming trans-synaptic nanocolumns to increase the efficacy of synapse transmission[6]. However, our understanding of the molecular mechanisms involved in trans-synaptic alignment is still limited.

[1]Interdisciplinary Institute for Neuroscience, Centre National de la Recherche Scientifique, Bordeaux, France. [2]Interdisciplinary Institute for Neuroscience, University of Bordeaux, Bordeaux, France. [3]University of Bordeaux, Bordeaux Proteome, Bordeaux, France. [4]VIB Center for Brain & Disease Research, Leuven, Belgium. [5]Department of Neurosciences, KU Leuven, Leuven Brain Institute, Leuven, Belgium. [6]These authors contributed equally: Malgorzata Lubas, Vasika Venugopal. ✉e-mail: ingrid.chamma@u-bordeaux.fr

Synaptic cell adhesion molecules (CAMs) are ideal candidates to regulate this process, inducing and organizing synapses through the formation of physical trans-synaptic signaling complexes[1,2]. Leucine-rich repeat transmembrane proteins (LRRTM1–4) are a family of synaptic CAMs present only in vertebrates and predominantly expressed in the brain; LRRTM2, the most synaptogenic isoform, exclusively localizes at excitatory postsynapses[7–10], where it exhibits low surface dynamics and forms compact nanoclusters[5]. LRRTM2 is a single-pass transmembrane protein with a large extracellular domain containing 10 leucine-rich-repeat (LRR) modules, a transmembrane segment, and a short cytoplasmic domain ending with a class I-like PDZ-binding motif ECEV, through which it binds PSD-95[8]. An intra-cellular motif YxxC (Y500/C503), 16 amino acids prior to the C-terminal end, is involved in the regulation of its expression and membrane stability[11,12], although the underlying mechanisms remain unclear. Through its extracellular domain, LRRTM2 binds pre-synaptic CAMs of the Neurexin family (1–3, α or β) (Nrxn) lacking a 30 amino acid insert at splice-site 4 (SS4-)[8,10,13]. Recently, the crystal structure of the LRRTM2–Nrxn1β complex revealed a glutamic acid residue at position 348 of LRRTM2 (E348) as critical for its calcium-dependent interaction with Nrxn1β[14]. These results contradict a previously hypothesized binding interface involving residues initially assigned as D260/T262[13,15] at position D259/T261 in the 9th extra-cellular LRR domain extensively investigated to study the physiolo-gical role of LRRTM2 binding to Nrxn[16,17]. Finally, LRRTM2 was shown to interact with AMPARs in heterologous cells[8,16], although whether this interaction occurs in neurons is uncertain.

Several studies investigated the role of LRRTM2 in synapse for-mation and function based on knock-down (KD)[8,16,18] or double knock-out (DKO) of LRRTM1-2[17,19], the main isoforms in CA1 hippo-campal neurons. As a result, the exact contribution of each isoform to synapse function is unclear. It is clear, however, that the physio-logical role of LRRTMs is tightly linked to AMPAR regulation, although the underlying mechanisms are not well understood. Double conditional knock-out (cKO) of LRRTM1-2 reduced AMPAR-mediated synaptic transmission and impaired long-term potentiation in CA1 pyramidal neurons[17], although whether LRRTM2 down-regulation alone is sufficient to decrease excitatory synapse density and AMPAR-mediated currents is controversial[8,20]. LRRTM2 KD or LRRTM1-2 double knock-down (DKD) decreased AMPAR synaptic levels[8,16], but the molecular mechanisms leading to this effect are unclear. LRRTM1-2 double cKO reduced the stability of over-expressed photo-activatable GFP-GluA1, suggesting that AMPARs might not be stabilized in spines in the absence of LRRTM1-2[17]. Finally, acute cleavage of an engineered LRRTM2 expressed on a KD background induced partial reorganization of synaptic AMPARs[21]. However, the molecular mechanisms by which this stabilization is achieved, and the interaction interfaces underlying them have remained elusive so far.

Here, we generated a cKO model of LRRTM2 to study the specific role of this isoform in synapse development, nano-organization, and function. We show that LRRTM2 cKO during synaptogenesis impairs excitatory synapse development and function. The C-terminal, but not the LRR domain, is responsible for synaptic clustering through the non-canonical PDZ-binding motif ECEV, and the YxxC motif maintains a low exocytosis rate. Once addressed at the plasma membrane, LRRTM2 stabilizes synaptic AMPARs through the recently identified Nrxn-binding interface containing E348, which also controls synapse formation, presynaptic RIM, and postsynaptic AMPAR nano-organization. These results demonstrate that LRRTM2 controls exci-tatory synapse organization and function through Nrxn binding site E348 providing insights into the molecular mechanism by which postsynaptic LRRTM2 trans-synaptically controls presynaptic nano-organization.

## Results

### Selective conditional knock-out of LRRTM2 impairs excitatory synapse formation and function

To specifically study the role of LRRTM2 in synapse formation, we generated a selective LRRTM2 cKO mouse model, where exon 2 was flanked by two loxP sites on a C57BL/6 background (Fig. S1a). The mouse LRRTM2 gene consists of 2 exons, with the first one covering part of the 5' UTR, the ATG translation initiation codon, and an additional nucleotide, while the protein-coding region resides in exon 2[7]. As expected from the conditional nature of the mutation, LRRTM2[Flox/Flox] mice developed normally and the mutation had no impact on their fertility, body weight, and feeding. To verify LRRTM2 gene inactivation upon Cre recombinase expression, we cultured hippocampal neurons from LRRTM2[Flox/Flox] mice and infected them with increasing concentrations of Cre recombinase-expressing lenti-viruses. LRRTM2 mRNA levels were assessed by reverse transcription quantitative PCR (RT-qPCR). Cre-recombinase expression dramatically reduced LRRTM2 mRNA levels in a dose-dependent manner confirm-ing gene inactivation (Fig. S1b). To quantify the impairment in protein levels, we performed Western blots on LRRTM2[Flox/Flox] primary cultures and organotypic hippocampal slices infected with Cre-recombinase lentiviruses. LRRTM2 protein was strongly reduced upon Cre-recombinase infection, confirming the cKO of LRRTM2 (Fig. S1c, d).

Because invalidation of LRRTM2 gene expression has led to con-flicting results in the literature[8,20], we re-examined the role of LRRTM2 in excitatory synapse formation and function here. Expression of Cre-recombinase in DIV7 LRRTM2[Flox/Flox] hippocampal neurons led to a ~30% reduction in excitatory synapse density in mature neurons (DIV15) (Fig. S1e, f), an effect rescued by co-expressing a biotin acceptor peptide (AP)-tagged LRRTM2 described in previous work[5]. Briefly, the AP tag was inserted at the N-terminal of LRRTM2, and AP-LRRTM2 was co-expressed with an endoplasmic reticulum (ER)-restricted biotin-ligase (BirA[ER]) that covalently adds biotin to the AP tag in the ER. Biotinylated LRRTM2 can thus be labeled at the cell surface using monomeric or tetrameric streptavidin[22,23] (Fig. 1a). Genetic invalidation of LRRTM2 did not affect inhibitory synapse development (Fig. S1g, h), demonstrating a specific effect of LRRTM2 in excitatory but not inhibitory synapse formation. Since LRRTM2 plays an impor-tant role in AMPAR-mediated synaptic transmission, we assessed sur-face AMPAR density upon invalidation of LRRTM2 expression and found a significant reduction of ~25% in GluA1/2 subunit surface levels (Fig. S1i, j). Finally, to assess the role of LRRTM2 cKO in synaptic transmission, we measured AMPAR miniature Excitatory Post Synaptic Currents (mEPSC) and found that AMPAR mEPCS frequency was strongly impaired in Cre-expressing cells, an effect rescued by co-expression of AP-LRRTM2 (Fig. S1k, l). Average mEPSC amplitude was not changed in Cre-expressing neurons, but a slight decrease was observed in the rescue condition, possibly due to mild overexpression in the recorded cells (Fig. S1l). However, LRRTM2 expression was not significantly increased compared to endogenous levels when assessed by immunolabeling on a larger cell population (Fig. S2). These results show that disruption of the synaptogenic protein LRRTM2 specifically impairs excitatory synapse formation and function.

### The C-terminal, but not the extracellular LRR domain, clusters LRRTM2 at excitatory synapses through the non-canonical PDZ-binding motif ECEV

Surface LRRTM2 is exclusively localized at excitatory postsynapses, where it forms compact clusters[5,8]. To examine the mechanisms of LRRTM2 regulation at the neuronal surface, we generated AP-tagged mutants of the C- and N- terminal domains of the protein (AP-LRRTM2-ΔC, AP-ΔLRR-LRRTM2) (Fig. 1b) and immunostained endogenous PSD-95 as a postsynaptic marker. In a previous study[12], we showed that the LRRTM2 C-terminal domain (CTD) was important for membrane

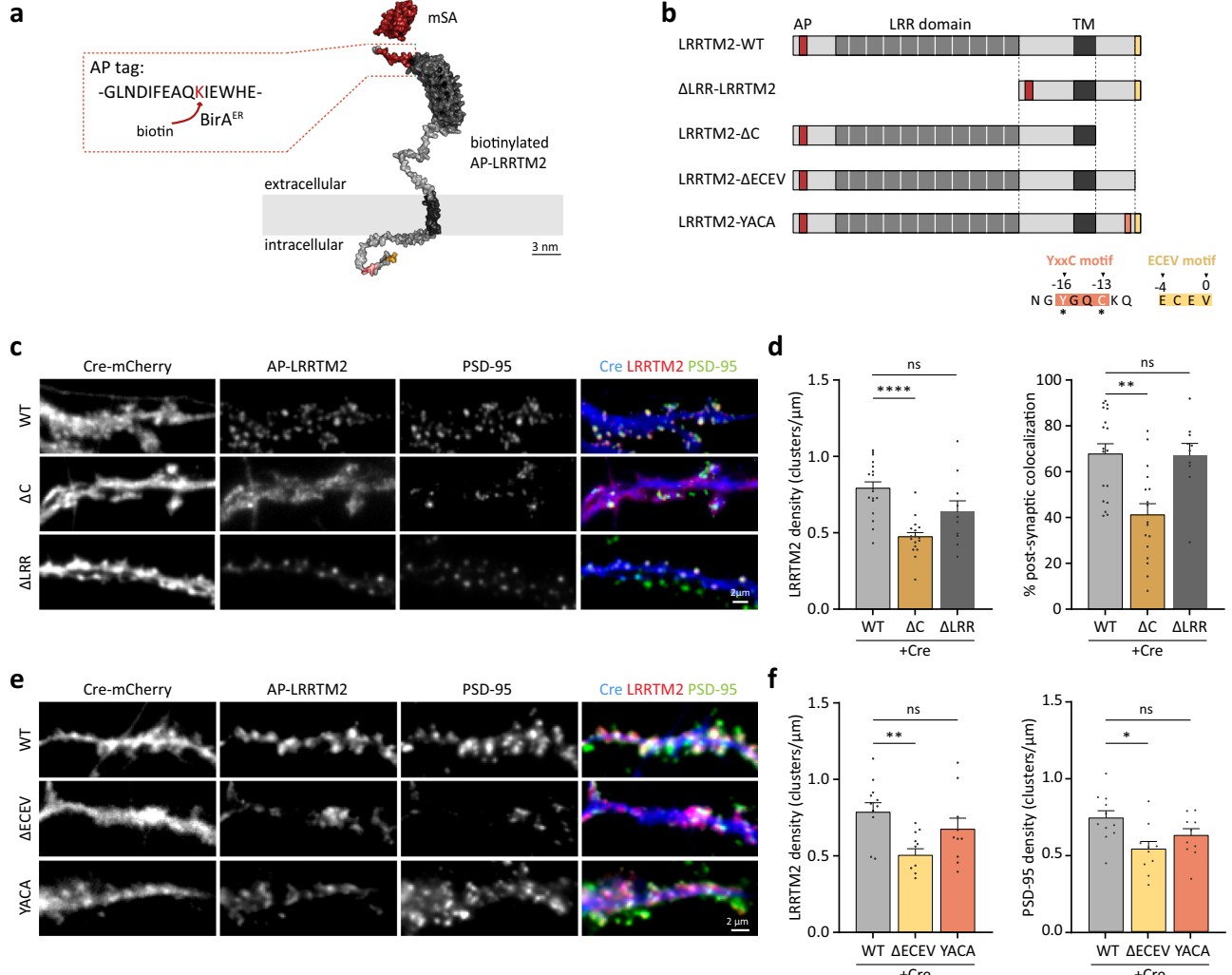

**Fig. 1 | The C-terminal, but not the extracellular LRR domain, clusters LRRTM2 at excitatory synapses through the non-canonical PDZ-binding motif ECEV.**
**a** Representation of LRRTM2 (AphaFold2 model using ColabFold v1.5.2[58]) at the plasma membrane carrying a biotin Acceptor Peptide (AP) tag (red) in its N-terminal domain and labeled with monomeric streptavidin (mSA, PDB4JNJ). **b** Schematics of LRRTM2- WT- and mutants lacking the extracellular LRR domain (ΔLRR-), the intracellular domain (-ΔC), the PDZ-binding domain ECEV (-ΔECEV), or mutated at the YxxC motif (-YACA). AP acceptor peptide, LRR leucine-rich repeat, TM transmembrane. Asterisks indicate the amino acids in the YxxC motif replaced with Alanines (LRRTM2-YACA). **c** DIV15 hippocampal neurons expressing Cre-mCherry, BirA[ER], biotinylated AP-LRRTM2 (WT, ΔC or ΔLRR) immunostained for endogenous PSD-95 as a postsynaptic marker. Cre-mCherry (blue) is overlaid with PSD-95 (green) and AP-LRRTM2 (red). **d** Quantification of AP-LRRTM2 (WT, ΔC, or ΔLRR) cluster density and percentage of PSD-95 clusters colocalized with AP-LRRTM2 clusters, showing decreased synaptic localization for the ΔC. Data acquired from three experiments were presented as mean values ± SEM (WT: $n = 18$, ΔC: $n = 17$, ΔLRR = 10 cells) **$p < 0.01$, ****$p < 0.0001$. Data were compared by one-way analysis of variance test, followed by post hoc Dunn's test. **e** DIV15 hippocampal neurons expressing Cre-mCherry, BirA[ER], biotinylated AP-LRRTM2 (WT, ΔECEV or YACA) immunostained for endogenous PSD-95. Cre-mCherry (blue) is overlaid with PSD-95 (green) and AP-LRRTM2 (red). **f** Quantification of AP-LRRTM2 (WT, ΔECEV or YACA) cluster density and PSD-95 cluster density showing decreased density for the ΔECEV, but not for the YACA. Data acquired from three experiments, presented as mean values ± SEM (WT: $n = 11$, ΔECEV: $n = 10$, YACA: $n = 10$ cells), *$p < 0.05$, **$p < 0.001$. Data were compared by one-way analysis of variance test, followed by post hoc Dunn's test.

stabilization and clustering using a knock-down strategy, but we had not explored the role of the extracellular LRR domain. Here, we confirm, using a cKO approach, that deletion of the CTD, but not the LRR domain, impairs LRRTM2 clustering and its localization at glutamatergic postsynapses (Fig. 1c, d). To further explore which motifs in the CTD are important for LRRTM2 synaptic clustering, we used AP-tagged mutants previously described[12]: deletion of the PDZ-like binding motif ECEV known to interact with the postsynaptic scaffolding protein PSD-95[8], and mutation of the YxxC sequence (Y500/C503 to alanines) known to regulate LRRTM2 membrane expression and diffusion[11,12] (AP-LRRTM2-ΔECEV and AP-LRRTM2-YACA). To assess that ΔC and ΔECEV mutants do not bind PSD-95, we expressed these in COS-7 cells with a GFP-tagged PSD-95 (Fig. S3). While PSD-95 homogeneously distributes in the cytoplasm of COS-7 cells in the absence of LRRTM2-WT, both proteins strongly co-cluster when expressed together (Fig. S3a). Using this assay as a readout of LRRTM2-PSD-95 interaction, we observed that deletion of the entire CTD or the ECEV motif alone completely disrupted PSD-95 cluster formation and hence interaction between the two proteins, while mutation of the YxxC motif did not affect PSD-95 clustering (Fig. S3a–c). In neurons, deletion of the ECEV motif disrupted LRRTM2 clustering to the same extent as deletion of the CTD, whereas mutation of the YxxC motif did not affect LRRTM2 clustering (Fig. 1e, f). Finally, while YxxC mutation did not affect PSD-95 cluster density, ECEV deletion induced a ~30% reduction (Fig. 1f), indicating that the effects on LRRTM2 and PSD-95 clustering require the ECEV motif, but not the YxxC motif.

## The ECEV motif retains LRRTM2 at excitatory postsynapses

In a previous study, we showed that the YxxC CTD motif is responsible for the confinement of LRRTM2[12]. However, due to incomplete knockdown of endogenous protein using shRNA, these effects could have been biased by the presence of endogenous protein. Thus, we re-examined the loss-of-function effect here, using cKO of LRRTM2. We tracked single LRRTM2 molecules using super-resolution imaging[24] and fluorophore-conjugated monomeric streptavidin (mSA) as previously described[5] in live hippocampal neurons from LRRTM2^Flox/Flox mice expressing Cre-recombinase, BirA^ER, postsynaptic marker Homer-1c-DsRed, and AP-LRRTM2 -WT, -ΔC, -ΔECEV, or -YACA (Fig. 2a, b). CTD deletion increased the overall mean square displacement (MSD), diffusion (D), and percentage of extra-synaptic trajectories, and reduced the fraction of immobile trajectories (Fig. 2c–f). YxxC mutation had similar effects, but ECEV deletion did not affect the overall diffusion, percentage of extra-synaptic tracks, or fraction of immobile trajectories (Fig. 2c–f). When restricting the analysis to synapses, we found that LRRTM2 diffusion was not affected by YxxC mutation or ECEV deletion, but the synaptic dwell time was strongly reduced upon CTD or ECEV deletion (Fig. 2g, h), indicating that the ECEV motif retains LRRTM2 at excitatory postsynapses, presumably through interactions with PSD-95. Only deletion of the entire CTD affected the fraction of immobile synaptic trajectories (Fig. 2i). These results indicate that CTD and ECEV deletion impaired stabilizing intracellular interactions at synapses (as shown by dwell time measurements, see ref. 25). In contrast, YxxC mutation might alter LRRTM2 diffusion as a result of increased surface expression[11,12] leading to saturation of available synaptic interaction sites[26,27].

## The YxxC motif regulates LRRTM2 membrane turnover maintaining a low exocytosis rate

To investigate the impact of YxxC mutation on membrane turnover and intracellular protein pools, and determine whether this motif impacts LRRTM2 exocytosis, we generated pH-sensitive super-ecliptic phluorin (SEP)-tagged- WT- and YACA-LRRTM2. When expressed on LRRTM2 cKO background, SEP-LRRTM2-WT and SEP-LRRTM2-YACA clustered at synapses and colocalized equally well with Homer1c-BFP (Figs. 3a and S4a–d). We first performed fluorescence recovery after photobleaching (FRAP) experiments to observe the turnover rate of LRRTM2 at synapses upon mutation of the YxxC motif (Fig. 3b–d). Fluorescence recovery of SEP-LRRTM2-WT measured within 12.5 min was low (<40%), in agreement with our previous observations using AP-LRRTM2-WT[5]. However, SEP-LRRTM2-YACA fluorescence recovery was drastically increased in spines leading to a strong reduction in the slow pool fraction (Fig. 3b–d), indicating faster replenishment of the protein at synapses. In shaft regions, the observed recovery was slightly higher, although the slow pool fraction remained unchanged (Fig. S4e, f). These results indicate that the YxxC motif controls LRRTM2 turnover at the plasma membrane. The increased recovery may be due to increased diffusion and membrane expression[11,12], and/or altered trafficking of the protein. We thus sought to examine what proportion of LRRTM2 was localized at the plasma membrane relative to intracellular pools, and whether this pool was altered upon YxxC mutation. We expressed SEP-LRRTM2-WT or -YACA on LRRTM2 cKO background (Fig. 3e) and applied a pH exchange protocol (Fig. 3f). The proportion of surface molecules can be determined by comparing fluorescence decrease in low extracellular pH with fluorescence increase caused by intracellular alkalinization[28]. We bath-applied a pH 5.5 solution to quench surface SEP-LRRTM2 fluorescence (~90%), followed by an ammonium chloride (NH₄Cl) solution to de-acidify intracellular vesicles by disrupting transmembrane proton gradients, leading to increased SEP fluorescence (~20% compared to baseline), which corresponds to LRRTM2 intracellular pools (Fig. 3g–i). Using this assay, we observed that the majority of LRRTM2 is localized at the plasma membrane (~80%), with only 20% of the protein in intracellular

compartments, and no change in these proportions upon YxxC mutation (Fig. 3h). Intracellular protein pools were observed inside spines as well as in dendritic regions, with no apparent change in these proportions upon YxxC mutation (Fig. 3j). These results suggest that the YxxC motif does not affect the localization or proportion of intracellular protein pools. We, therefore, hypothesized that increased membrane expression and diffusion of LRRTM2-YACA might be due to increased protein exocytosis. To test this in a simplified system, we expressed SEP-LRRTM2-WT or -YACA in COS-7 cells (Fig. S5). To isolate individual exocytosis events, we photobleached the whole cell surface. Intracellular SEP fluorescence is quenched under basal conditions, and therefore, only the membrane fluorescence was photobleached (Fig. S5a), allowing imaging and precise localization of individual exocytic events (Fig. S5b). Using this assay, we found that the frequency of exocytic events was increased ~3-fold in SEP-LRRTM2-YACA compared to WT (Fig. S5c), and the decay of exocytic events was drastically decreased (Fig. S5d), suggesting that LRRTM2 is released faster upon YxxC mutation, and that released molecules diffuse faster away from exocytosis sites. Thus, the YxxC motif regulates LRRTM2 membrane turnover by maintaining a low exocytosis rate.

## LRRTM2 controls synaptic AMPAR stabilization through extracellular coupling

Once addressed at the plasma membrane and stabilized at excitatory synapses, we asked how LRRTM2 influences AMPAR function, as the underlying mechanisms remain unclear[8,16,21]. We first immunoprecipitated surface LRRTM2 in hippocampal neurons and performed mass spectrometry analysis. In this context, we detected the GluA2 subunit of AMPARs along with all three Nrxn isoforms (Nrxn1–3) and PSD-95, indicating that AMPARs are present in the surface proteome of LRRTM2 in neurons (Fig. S6a, b and Table S1). To further investigate the interaction between LRRTM2 and AMPARs, we performed pull-down assays in COS-7 cells expressing AP-LRRTM2 and SEP-GluA2. WT-LRRTM2, but not ΔLRR-LRRTM2, co-precipitated with GluA2, indicating that the extracellular domain of LRRTM2 is necessary for LRRTM2 association with GluA2[8] (Fig. S6c). In parallel, we performed live co-recruitment assays in cells to determine whether GluA2 and LRRTM2 co-segregate in a cellular context. We cross-linked SEP-GluA2 receptors at the surface of COS-7 cells and examined the co-aggregation of LRRTM2. WT-LRRTM2 efficiently co-segregated with cross-linked GluA2 in these experiments, as seen by the formation of colocalized clusters (Fig. S6d). In contrast, ΔLRR-LRRTM2 failed to co-segregate with GluA2 (Fig. S6d). These experiments indicate that GluA2 and LRRTM2 physically associate and are part of the same molecular complexes in neurons as well as in heterologous cells. To evaluate the stabilization effect of LRRTM2 on AMPAR surface expression, we performed live imaging experiments of GluA1 (Fig. 4a–d) and GluA2 (Fig. 4e–h) subunit trafficking at excitatory synapses using SEP-tagged AMPAR subunits to exclusively image the surface pools of AMPARs. FRAP was performed in the presence of endogenous LRRTM2 or on a cKO background. Selective disruption of LRRTM2 expression led to increased recovery of surface GluA1 and GluA2 at synapses (Fig. 4b, c, f, g), and to a reduction in the slow pool fractions of both subunits (Fig. 4d, h). These effects were rescued by the re-expression of AP-LRRTM2-WT, showing that LRRTM2 stabilizes synaptic AMPARs by reducing their surface turnover. Previous studies showed that LRRTM2 directly interacts with AMPARs in heterologous cells through the extracellular LRR domain[8] and that this domain is involved in synapse physiology[16,17]. However, the stabilizing effect of LRRTM2 on AMPARs could also be indirect through altering post-synaptic scaffold density, as shown for other adhesion proteins[29]. To discriminate between these possibilities, we assessed how CTD or LRR domain deletions (AP-LRRTM2-ΔC and AP-ΔLRR-LRRTM2) affect the turn-over of AMPARs using FRAP (Fig. S7). LRR domain deletion led to a strong reduction in SEP-GluA1 intensity (Fig. S7a, b) and increased

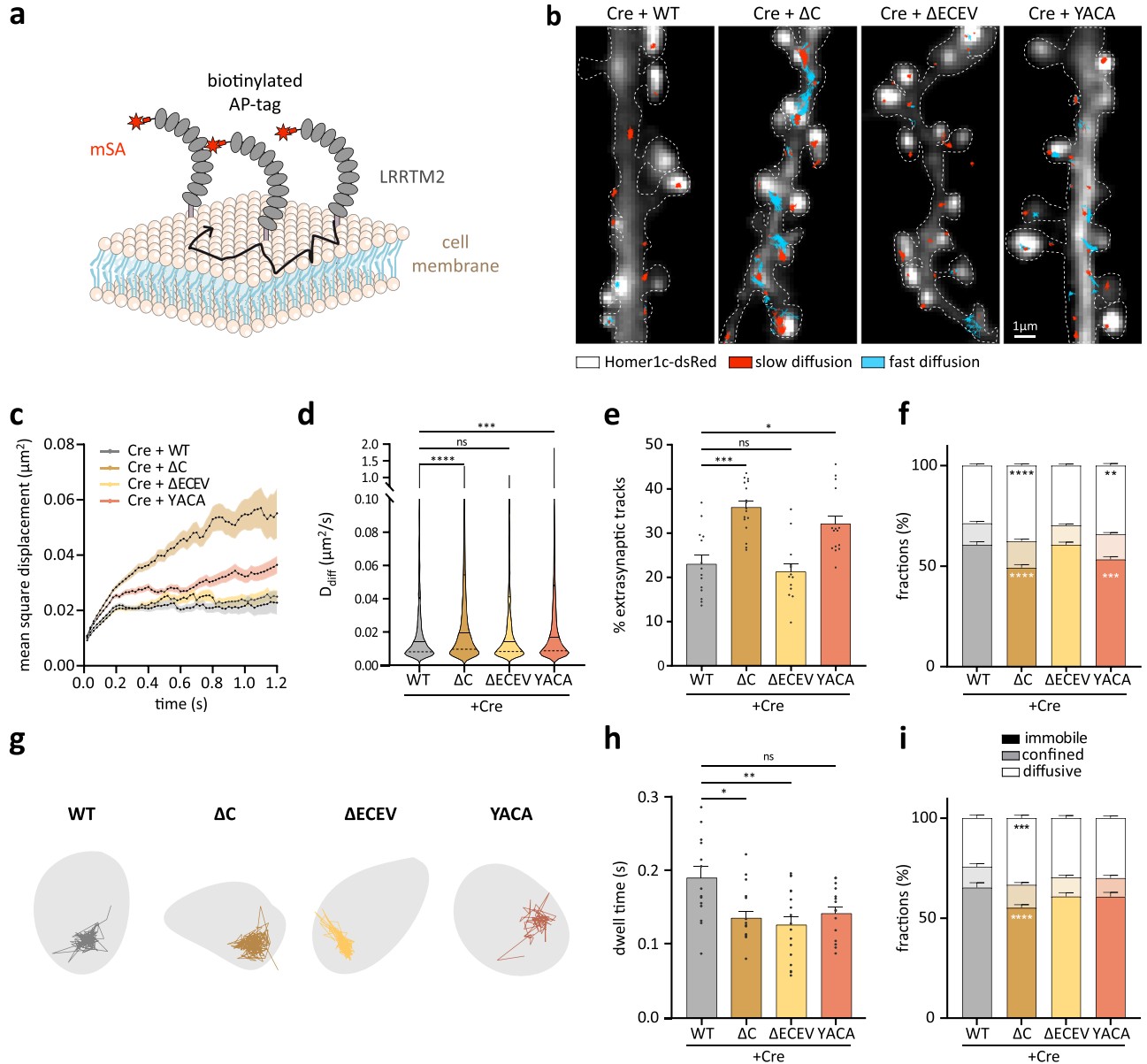

**Fig. 2 | Membrane diffusion and confinement of LRRTM2 are regulated by the C-terminal domain. a** Schematics of mSA-labeled AP-LRRTM2 diffusing at the plasma membrane. **b** Representative images from DIV15 neurons co-expressing Cre-EGFP, Homer1c-DsRed, BirA[ER], and AP-LRRTM2 mutants (WT, ΔC, ΔECEV or YACA) labeled with mSA-STAR935P to track individual molecules by universal Point Accumulation for Imaging in Nanoscale Topography[24] (uPAINT). Homer1c-DsRed (gray) overlaid with AP-LRRTM2 trajectories (immobile: $D < 0.005 \, \mu m^2 \, s^{-1}$, red; diffusive: $D > 0.005 \, \mu m^2 \, s^{-1}$, cyan; see the "Methods" section). **c** Mean square displacement over time, showing increased overall diffusion upon deletion of CTD, or mutation of YxxC. Data presented as mean values ± SEM. **d** Diffusion coefficients of free-diffusing proteins ($D_{diff}$, see the "Methods" section) shown as violin plots for AP-LRRTM2-WT, -ΔC, -ΔECEV and -YACA (trajectories, WT: 1779, ΔC:2671, ΔECEV: 2070, YACA: 2918; ***$p < 0.001$, ****$p < 0.0001$). Data was compared by one-way analysis of variance test, followed by post-hoc Dunn's test. **e** Percentage of non-synaptic tracks, showing an increase with ΔC and YACA-LRRTM2 (*$p < 0.05$, ***$p < 0.001$). Data presented as mean values ± SEM. Data were compared by one-

way analysis of variance test, followed by post-hoc Dunn's test. **f** Fractions of immobile, confined, and diffusive trajectories of AP-LRRTM2 mutants, showing decreased immobile fractions for the ΔC and the YACA mutants (**$p < 0.01$, ***$p < 0.001$, ****$p < 0.0001$). Data presented as mean values ± SEM. Data were compared by two-way analysis of variance test, followed by Tukey's multiple comparison test. **g** Individual synaptic trajectories overlaid with Homer1c-DsRed (gray). **h** Mean synaptic dwell time of AP-LRRTM2, showing a specific decrease of time spent at synapses in the absence of the PDZ-like binding motif (ΔC, *$p < 0.05$ and ΔECEV, **$p < 0.01$), but not in YACA condition. Data presented as mean values ± SEM. Data were compared by one-way analysis of variance test, followed by post-hoc Dunn's test. **i** Fractions of immobile, confined, and diffusive trajectories for synaptic AP-LRRTM2 showing a decrease in immobile fraction in ΔC condition (***$p < 0.001$, ****$p < 0.0001$). Data presented as mean values ± SEM. Data were acquired from three independent experiments (WT: $n = 16$, ΔC: $n = 18$; ΔECEV: $n = 18$, YACA: $n = 16$ cells). Data were compared by two-way analysis of variance test, followed by Tukey's multiple comparison test.

turnover of synaptic SEP-GluA1 (Fig. S7c, d), consistent with previous work showing that the LRR domain is critical for AMPAR stabilization by LRRTM2[8,16]. CTD deletion however, led to reduced SEP-GluA1 intensity (Fig. S7a, b), but only slight increase in fluorescence recovery of SEP-GluA1 (Fig. S7c, d), an effect that may be attributed to the

observed reduction in PSD-95 recruitment, critical for AMPARs stabilization at synapses[30,31]. Importantly, only LRR deletion significantly reduced the slow pool fraction of synaptic AMPARs, while CTD deletion had no effect (Fig. S7e). Finally, these effects were selective to synapses, as turnover of shaft SEP-GluA1 was comparable in all

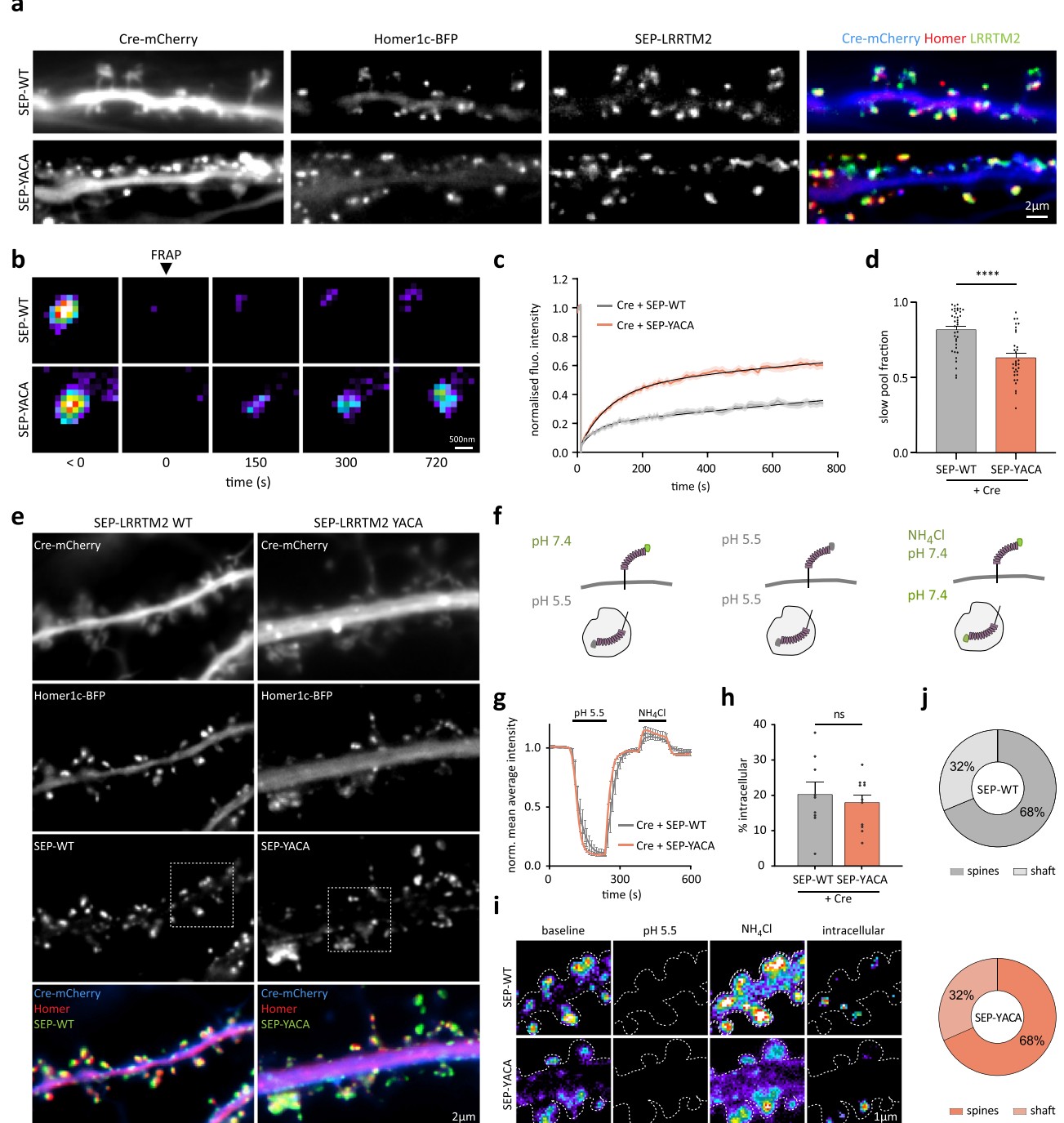

**Fig. 3 | The YxxC motif regulates LRRTM2 membrane turnover. a** Representative images of DIV15 hippocampal neurons expressing Cre-mCherry, Homer1c-BFP and either SEP-LRRTM2-WT or SEP-LRRTM2-YACA. On the right, Cre-mCherry (blue) is overlaid with Homer (red) and LRRTM2 (green). **b** Pseudocolor images of SEP fluorescence recovery after photobleaching in spines from neurons expressing Cre-mCherry and SEP-LRRTM2-WT or -YACA. **c** Corresponding normalized fluorescence recovery curves in spines and **d** slow pool fraction in both conditions. Data are presented as mean values ± SEM. Data acquired from three independent experiments (SEP-LRRTM2-WT: $n = 37$ regions, SEP-LRRTM2-YACA: $n = 32$ regions; ****$p < 0.0001$). Data were compared by non-parametric Mann–Whitney test. **e** Representative images of DIV15 hippocampal neurons expressing Cre-mCherry (blue), Homer1c-BFP (red), and SEP-LRRTM2-WT or -YACA (green). Dashed squares indicate the positions of panel **i**. **f** Schematics of the pH change protocol used to visualize intracellular and extracellular protein pools. At pH 7.4, the SEP-tag is

fluorescent, whereas at acidic pH (pH 5.5), its fluorescence is quenched. Bath application of pH 5.5 renders surface SEP-tagged proteins non-fluorescent, and $NH_4Cl$ de-acidifies intracellular vesicles, rendering intracellular SEP-tagged proteins fluorescent. **g** Normalized mean average intensity of SEP-LRRTM2-WT and SEP-LRRTM2-YACA overtime during the pH change protocol illustrated in f. Data are presented as mean values ± SEM. **h** Percentage of SEP-LRRTM2-WT and SEP-LRRTM2-YACA protein fluorescence signal localized in the intracellular compartments of dendrites. Data are presented as mean values ± SEM. Data were compared by non-parametric Mann–Whitney test. **i** Pseudocolor images of insets from (**e**) during the pH change protocol, revealing intracellular protein pools by subtraction of the "baseline" signal (extracellular) from the "$NH_4Cl$" signal (total) in the shaft and spines. **j** Percentage of SEP-LRRTM2-WT and SEP-LRRTM2-YACA intracellular pool distribution in spines and dendritic shaft. **g–i** Data acquired from three independent experiments (SEP-WT: $n = 9$ and SEP-YACA: $n = 11$ cells).

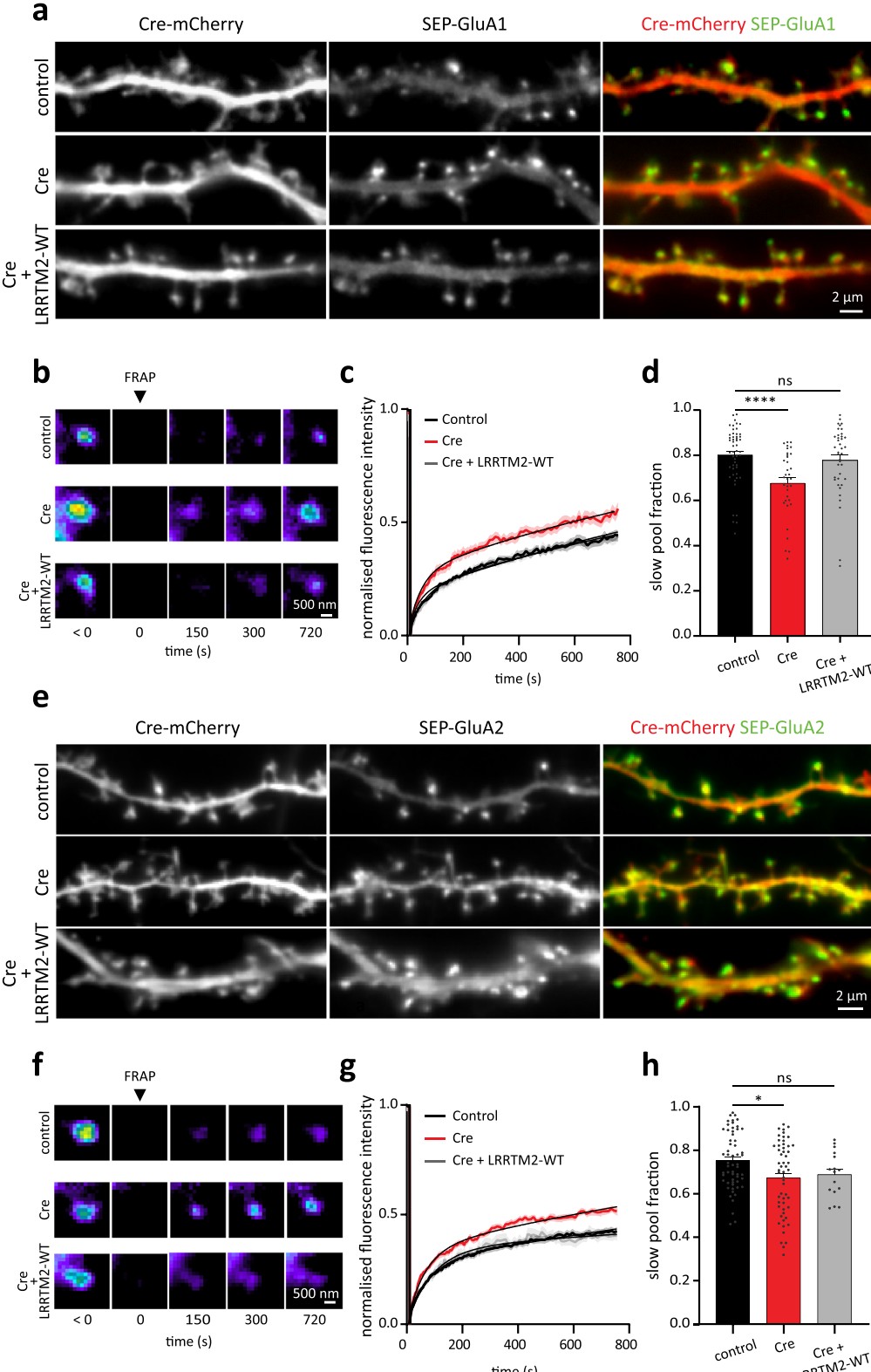

conditions (Fig. S7f). Thus, LRRTM2 stabilizes synaptic AMPARs through extracellular interactions.

### Stabilization of synaptic AMPARs requires Neurexin-binding interface containing E348

AMPAR destabilization upon LRR domain deletion was previously attributed to the binding of Nrxns to LRRTM2, since point mutations designed to impair Nrxn binding in the 9th LRR motif of LRRTM2 (D259, T261)[13] affected AMPAR-dependent synaptic transmission[16,17]. However, a recent crystallography study mapped the binding interface between Nrxn1β and LRRTM2 to the C-terminal cap of the extracellular LRR domain, where they identified the residue E348 as critical for calcium-mediated Nrxn1β/LRRTM2 interaction[14], questioning previous observations. To examine which of these residues

**Fig. 4 | LRRTM2 cKO impairs synaptic AMPAR stabilization. a** DIV15 hippocampal neurons expressing soluble mCherry and SEP-GluA1 subunit of AMPARs (control), Cre-mCherry and SEP-GluA1 (Cre) or mCherry, SEP-GluA1, BirA[ER] and AP-LRRTM2-WT (Cre + LRRTM2-WT). On the right, mCherry (red) is overlaid with SEP-GluA1 (green). **b** Pseudocolor images of FRAP experiments performed on SEP-GluA1 localized in spines for the different conditions described in (**a**). **c** Corresponding normalized fluorescence recovery curves, showing faster recovery of SEP-GluA1 fluorescence in the absence of LRRTM2, and **d** slow pool fraction, showing a decrease of this fraction in the absence of LRRTM2. Data are presented as mean values ± SEM. Data obtained from three independent experiments (control: $n = 51$; Cre: n = 34; Cre + LRRTM2-WT: 39 regions, ****$p < 0.0001$). Data were compared by one-way analysis of variance test, followed by post hoc Dunn's test. **e** DIV15

hippocampal neurons expressing soluble mCherry and SEP-GluA2 subunit of AMPARs (control), Cre-mCherry and SEP-GluA2 (Cre) or mCherry, SEP-GluA2, BirA[ER] and AP-LRRTM2-WT (Cre + LRRTM2-WT). On the right, mCherry (red) is overlaid with SEP-GluA2 (green). **f** FRAP of SEP-GluA2 localized in spines for the different conditions described in (**e**), showing faster recovery in the absence of LRRTM2. **g** Corresponding normalized fluorescence recovery curves and **h** slow pool fraction show that GluA2 is less stabilized at synapses in the absence of LRRTM2. Data are presented as mean values ± SEM. Data obtained from three independent experiments (control: $n = 58$; Cre: $n = 55$; Cre + LRRTM2-WT: $n = 16$ regions, *$p < 0.05$). Data were compared by one-way analysis of variance test, followed by post hoc Dunn's test.

are important for Nrxn binding and AMPAR stabilization, we generated mutants of LRRTM2 extracellular domain: AP-EQ-LRRTM2, containing the E348Q substitution that abolishes Nrxn1β binding in vitro[14], and AP-DT/AA-LRRTM2, containing the double mutation D259A/T261A in the 9th LRR domain previously described[13] (Fig. 5a, b). We expressed these mutants with BirA[ER] in heterologous COS-7 cells that do not express known Nrxn partners and performed Nrxn-binding assay using purified Nrxn1β (-SS4)[32,33]. All mutants were expressed at the COS-7 cell surface, as observed using live surface labeling with mSA (Fig. 5c), although DT/AA mutant expression was decreased by 50% compared to AP-LRRTM2-WT and AP-EQ-LRRTM2 (Fig. 5d). Nrxn1β-Fc bound to all cells expressing AP-LRRTM2-WT, but EQ mutation completely abolished Nrxn1β-Fc binding, while the DT/AA mutation only did so by 50% (Fig. 5e). These results show that D259 and T261 mutations do not completely impair Nrxn-binding and confirm that Nrxn-binding involves the critical E348 in LRRTM2 identified by Yamagata and colleagues[14]. To further confirm these results and examine the role of these mutants in synapse formation, we performed co-culture assays. As a synaptogenic protein, LRRTM2 induces presynapse formation through binding to Nrxns when expressed in non-neuronal cells[8,9]. To examine whether EQ and DT/AA mutants were still able to induce synapse formation in wild-type neurons, we expressed these mutants in COS-7 cells and co-cultured them with primary hippocampal neurons (Fig. 5f). As expected, accumulation of presynaptic synapsin-1 onto COS-7 cells expressing WT-LRRTM2 was abolished in the presence of the mutant lacking the entire extra-cellular LRR domain (Fig. 5f, g). EQ mutation abolished the recruitment of synapsin-1 to the same extent as deletion of the entire LRR domain, whereas DT/AA mutation had no effect on synapsin-1 accumulation (Fig. 5g). These results demonstrate that EQ mutation disrupts Nrxn-binding and prevents synaptogenesis in cocultures. Finally, we measured synapse density in neurons expressing these mutants and observed a selective decrease in the EQ but not DT/AA mutant (Fig. 5h, i). Thus, we conclude that E348 is necessary for Nrxn binding and synapse formation.

We next sought to determine whether this binding interface played a role in AMPAR stabilization. In neurons, unlike COS-7, both mutants expressed at the cell surface to similar levels compared to LRRTM2-WT (Figs. 5j and S8a, b). Interestingly, EQ mutation significantly reduced SEP-GluA1 intensity selectively in spines, while DT/AA mutant had no effect (Fig. 5k). FRAP on synaptic SEP-GluA1 in cKO hippocampal neurons expressing AP-LRRTM2 mutants showed that both EQ and DT/AA mutants destabilized surface AMPARs recovery at synapses and that this destabilization is similar to that caused by deletion of the whole LRR domain (ΔLRR) (Figs. 5l, m and S7d). However, only the EQ mutation reduced significantly the slow pool fraction of AMPARs (Fig. 5n), indicating a selective effect of this mutant on the stabilization of synaptic AMPARs. Recovery of shaft SEP-GluA1 was not affected by the mutations (Fig. S8c, d), supporting our findings that LRRTM2 selectively stabilizes AMPARs at synapses.

## LRRTM2−Neurexin binding interface containing E348 is critical for presynapse nano-organization and AMPAR sub-positioning at synapses

To examine how LRRTM2 mutations affect presynapse nano-organization and postsynaptic AMPAR positioning, we performed dual-color super-resolution direct STochastic Optical Reconstruction Microscopy (dSTORM). We immunolabeled the presynaptic active zone scaffold proteins RIM1/2 and the postsynaptic surface pool of GluA1/2 subunits of AMPARs, two main components of trans-synaptic nanocolumns[6] (Fig. 6a) and examined their nanoscale organization in the presence of EQ- or DT/AA- LRRTM2 mutants. In the WT condition, RIM1/2 and GluA1/2 formed compact subsynaptic domains (SSDs), as previously described[3,4,6,34]. At the presynaptic level, only the Nrxn-binding interface mutant (EQ) induced a disruption in RIM nanocluster size and content (Fig. 6b, c), showing reduced localization and reduced SSD surface, whereas DT/AA mutant did not affect presynapse organization (Fig. 6b, c). These results show that trans-synaptic interaction between LRRTM2 and Nrxns is important for presynaptic RIM nano-organization. At the postsynaptic level, endogenous AMPARs formed SSDs of ~80 nm diameter facing presynaptic RIM SSDs (Fig. 6b). Mutation of the Nrxn binding interface E348 induced a strong impairment in AMPARs nano-organization, reducing SSD surface and content (Fig. 6b, d). A similar effect was observed in DT/AA mutant (Fig. 6b, d), indicating that both interfaces are required for proper nano-organization of postsynaptic AMPARs. Interestingly, the RIM1/2 SSD number per synapse was reduced in the EQ mutant but not in the DT/AA, whereas GluA1/2 SSD numbers were reduced with both mutants (Fig. 6e). Finally, the centroid-to-centroid distances between RIM1/2 and GluA1/2 SSDs were increased in the presence of both EQ- and DT/AA-LRRTM2 mutants compared to the WT (Fig. 6f). These results show that LRRTM2 binding to Nrxns through E348 is critical for presynaptic RIM nano-organization and postsynaptic AMPAR subdomain organization, whereas the 9th LRR domain plays a role in AMPAR nanoscale organization, but not in presynapse organization.

To address the functional effects of these mutations in excitatory synaptic transmission, we recorded mEPSCs. Surprisingly, we found that while the average mEPSCs amplitude was not altered by EQ or DT/AA mutations, the distribution of individual amplitudes was significantly shifted towards higher values (Fig. 6g, h), suggesting a non-uniform effect on individual events that are not detected when averaging the whole population. In particular, the fraction of events with smaller amplitudes (<15 pA) was strongly reduced in the mutant conditions compared to the WT, suggesting a selective loss of weak synapses. In agreement, the distribution of mEPSC inter-event intervals in the EQ condition was shifted to higher values compared to the WT condition (Fig. 6i), together with a slight but not significant decrease in average mEPSCs frequency. In contrast, we observed a shift towards lower values in the DT/AA condition despite unchanged average frequency (Fig. 6i). Such a differential effect of the mutants on mEPSCs frequency could reflect a different impact on presynaptic organization and function.

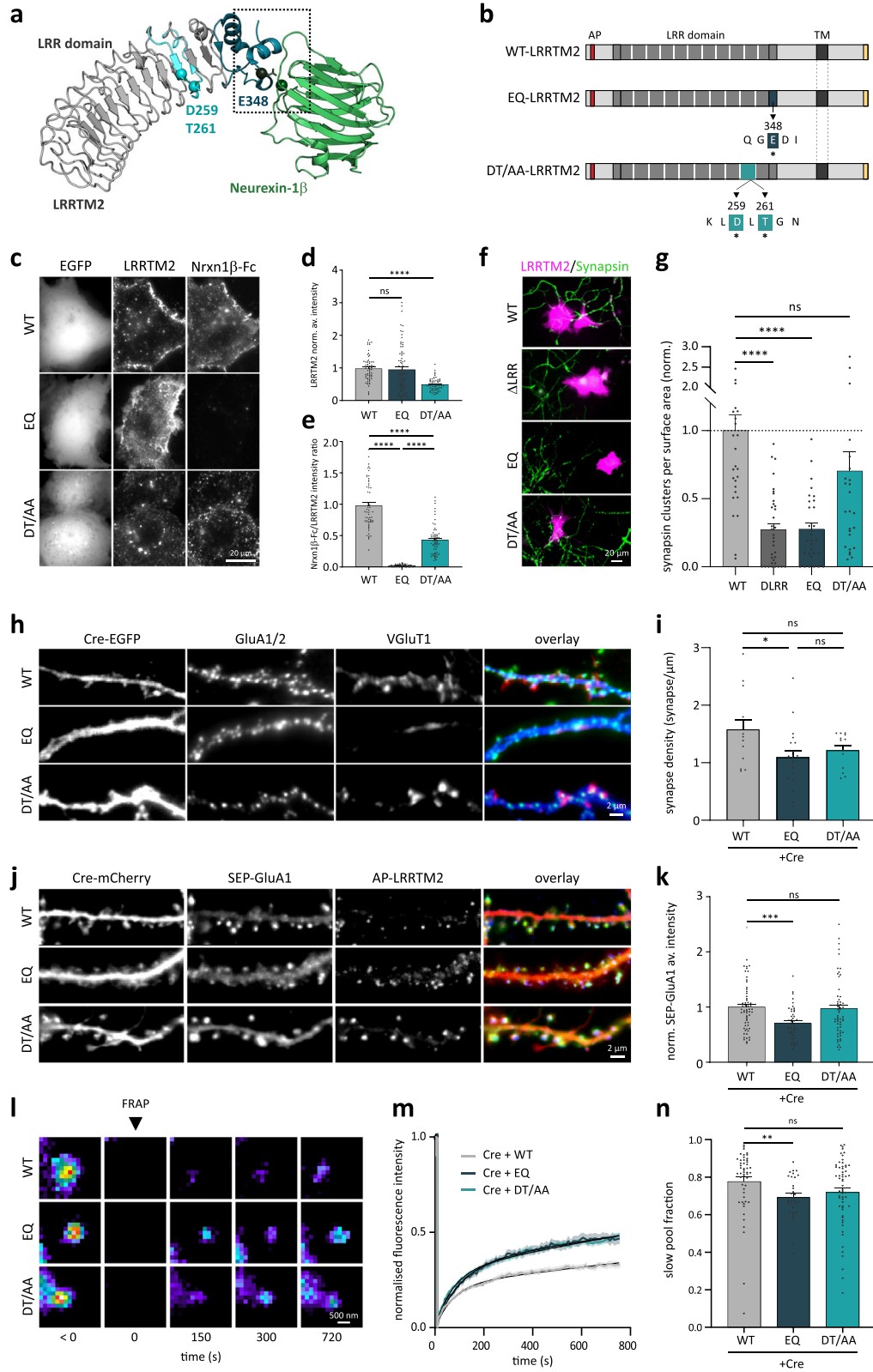

## Discussion

To our knowledge, the role of Nrxn-binding interface[14] involved in Nrx−LRRTM2 interaction has never been addressed in synapse organization and function. We demonstrate here, using a conditional knock-out model, that LRRTM2 is critical for the formation and function of excitatory synapses, with no apparent compensation by other LRRTM isoforms and no effect on inhibitory synapses. We show that the CTD is responsible for synaptic confinement at synapses and low membrane turnover, through selective intracellular motifs. Conversely, the N-terminal domain (NTD) is critical for synapse formation through binding to Neurexins and is necessary for finely positioning key pre- and post-synaptic components at the nanoscale. Specifically, the Nrxn-binding interface in LRRTM2 containing E348 recently identified[14] is critical for AMPAR synaptic turnover and for nanoscale

**Fig. 5 | LRRTM2 controls synaptic AMPAR stabilization through neurexin-binding site. a** Crystal structure of hLRRTM2 (gray) in complex with Neurexin-1β (green) (PDB 5Z8Y), showing interaction site E348 recently identified[14] (dark blue) and calcium ion (green), and D259 and D261[15] (light blue). **b** Schematics of EQ- (E348 mutated to Glutamine (Q)) and DT/AA- LRRTM2 (D259/T261 mutated to Alanines (A)). **c** COS-7 cells expressing EGFP and biotinylated WT-, EQ-, or DT/AA-LRRTM2, labeled with mSA-ATTO565, incubated with Nrxn1β-Fc and antiFc-A647. No Nrxn1β-Fc at EQ-LRRTM2 cell surface, showing disrupted binding. **d** Normalized average surface intensity of LRRTM2. **e** Average surface intensity of Nrxn1β-Fc normalized to expression levels of AP-LRRTM2, showing total disruption of Nrxn1β-binding exclusively with EQ. Data from three independent experiments (cells, WT: $n = 60$, EQ: $n = 77$, DT/AA: $n = 62$) ****$p < 0.0001$. **f** Co-culture showing recruitment of endogenous presynaptic synapsin1 (green) onto COS-7 cells expressing WT-LRRTM2 or DT/AA-LRRTM2, but not ΔLRR- or EQ-LRRTM2 (magenta) and **g** corresponding quantifications showing loss of synapsin1 recruitment in absence of LRR domain or mutation of E348, but not mutation of D259/T261. Data from three independent experiments (cells, WT: $n = 29$, ΔLRR: $n = 35$, EQ: $n = 31$, DT/AA: $n = 27$, ****$p < 0.0001$). **h** DIV15 neurons expressing Cre-EGFP, BirA$^{ER}$, and WT-, EQ- or DT/AA-LRRTM2 labeled for endogenous GluA1/2 and VGluT1 and **i** corresponding quantifications of synapse density (GluA1/2/VGluT1 apposition), showing specific decreased density upon EQ mutation. Data from three experiments (cells, WT: $n = 14$, EQ: $n = 20$, DT/AA: $n = 13$) *$p < 0.05$. **j** DIV15 neurons expressing Cre-mCherry, SEP-GluA1, BirA$^{ER}$ and WT-, EQ- or DT/AA-AP-LRRTM2. **k** Normalized average intensity of spine SEP-GluA1, showing decreased intensity with EQ-LRRTM2 (regions, WT: $n = 60$, EQ: $n = 45$, DT/AA: $n = 57$), ***$p < 0.001$. **l** SEP-GluA1-containing spines before and after FRAP, **m** normalized FRAP curves in spines and **n** corresponding slow pool fraction, showing a selective reduction in EQ-LRRTM2. Data from three independent experiments (regions, WT: $n = 50$, EQ: $n = 31$, DT/AA: $n = 59$), **$p < 0.01$. **d**, **e**, **g**, **i**, **k**, **n** Data presented as mean values ± SEM, compared by one-way analysis of variance test, followed by post hoc Dunn's test.

organization of both AMPARs and the active zone protein RIM that mediates vesicle priming at presynaptic terminals (Fig. 7). In contrast, the previously identified residues D259/T261 in the 9th LRR domain[13,15] exert a function specific to the postsynapse, by affecting AMPAR nano-organization, without effects on presynapses. These results indicate that while only E348 is necessary for Nrxn binding and synapse formation, the D259/T261-containing interface could extend the stabilization effects on AMPARs, possibly through Nrxn-independent interactions occurring at this concave interface, and that these interactions are not incompatible with Nrxn binding. There is a substantial discrepancy concerning the role of LRRTM2 in synapse formation and function originating from data obtained in knock-down experiments in hippocampal cultures[8,18,20]. The first KD study showed impaired synapse formation[8], but a later study using DKD of LRRTM1-2 showed no effects on synapse numbers[20]. Then, a triple KD of Neuroligin3, LRRTM1, and LRRTM2 on Neuroligin1 KO background showed impaired synapse development[18]. Since a cDKO for LRRTM1-2 showed no effect on synapse density but impairments in AMPAR-mediated transmission[17]. Finally, an LRRTM1-2 DKO model showed a reduction in excitatory synapses and mEPSC frequency but not amplitude in CA1 neurons[19]. These controversies could be due to double or triple KD or constitutive KO that can lead to compensation mechanisms. Thus, selective conditional KO of a unique protein isoform remains the best approach to studying the specific role of a given protein. Our results are consistent with selective effects of LRRTM2 at excitatory synapses and downregulation of AMPAR function and surface expression[8,19].

Some studies have inferred that LRRTM2 binding to Nrxns is responsible for excitatory synapse function and long-term potentiation[16,17]. However, these were performed using mutations (assigned as D260/T262) initially thought to disrupt LRRTM2–Nrxn binding, shown first with Nrx-binding assay[13], and later by crystallography using an engineered thermostabilized LRRTM2, where 33% of the residues were mutated[15]. Recently, however, this model was questioned by a new report on the crystal structure of the Nrxn1β–LRRTM2 complex[14]. This new study determined the structure of native LRRTM2 and showed that the interaction interface between Nrxn1β and LRRTM2 involves a glutamic acid residue in position 348 (E348) distant from the LRR9 domain. The authors also observed secretion defects in the DT/AA mutant and concluded that mutating D259 and T261 residues might affect protein folding, thereby disturbing to a certain extent Nrxn-binding, but that the 9th LRR domain was mostly not involved in Nrxn interaction[14]. Our results using purified Nrxn1β-binding assay in COS-7 cells show that only EQ mutation completely abolishes Nrxn1β binding. We also observed defects in membrane expression of the DT/AA mutant, specifically in COS-7 cells but not in neurons. In our experiments, the DT/AA mutant was still able to bind Nrxn1β, although at lower levels compared to LRRTM2-WT. In co-culture assays, DT/AA mutation did not impair presynapse

formation in contacting axons, whereas deletion of the LRR domain or single residue mutation EQ disrupted Nrxn-dependent synaptogenic properties of LRRTM2. Finally, only EQ mutation reduced synapse density in cultured neurons. Thus, we conclude that E348−but not D259/T261−is critical for Nrxn binding and synapse formation.

While EQ mutant strongly affected synapse numbers, AMPAR stabilization, and nano-organization, these effects were not paralleled by significant functional alterations of average mEPSCs as observed in LRRTM2 cKO (Fig. S1, see also ref. 19). mEPSCs are highly sensitive to synaptic parameters such as glutamate release probability, AMPAR content or number of active synapses, and mEPSC average amplitude, like evoked quantal events, is decreased when AMPARs SSDs are spatially shifted from presynaptic release sites[29]. A possibility is that the functional impact of LRRTM2 mutants may be restricted to sub-populations of synapses or subsynaptic nanocolumns of different molecular composition, as described in cerebellar cortex with Nrxn splice variants[35], it may thus be difficult to detect these changes when recording mEPSCs from the cell body. Refining our analysis, we found the cumulative mEPSC amplitude distributions of the mutants shifted towards higher values compared to WT. While the increase in amplitudes appears at odds with the decrease in AMPARs SSD density and size, this might be explained by the selective loss of small events (<15 pA) observed with the mutants. This is supported by the decrease in the number of synapses and GluA1 SSDs per synapse found with EQ mutant and by the trend towards larger inter-event intervals compared to WT. In contrast to the EQ, DT/AA mutant slightly decreased the inter-event intervals, although synapse numbers and average mEPSC frequency were not affected. Thus, EQ and DT/AA mutants may have differential effects on presynapse organization and function.

Using super-resolution imaging, we show that only EQ mutation−but not DT/AA−affects presynaptic nano-organization. This mutant fails to bind Nrxn, fails to recruit synapses, reduces synapse density, and disrupts presynaptic RIM nano-organization, AMPAR synaptic turnover, and AMPAR nanoscale positioning at synapses. Consistent with impaired Nrxn-binding, genetic ablation of Nrxn3 in hippocampal neurons showed a similar phenotype with reduced GluA1 SSD per synapse, and decreased AMPAR-RIM trans-synaptic alignment[36]. How Nrxns are linked to RIM nano-organization remains elusive, but a recent study showed that liprin-α interacts with RIM and clusters Nrxns through CASK, leading to presynapse recruitment and assembly[37]. Consistently, our results show that LRRTM2 trans-synaptically controls presynapse assembly via binding to Nrxns through E348. In contrast, the DT/AA mutant only affected AMPAR stabilization and nano-organization but not presynapse assembly, consistent with the fact that it does not impair Nrxn-binding or synapse formation. How DT/AA mutations affect AMPAR nano-organization and what interactions are mediated through this interface is not clear. Since LRRTM2 binds Nrxns through its C-terminal cap, the concave interface remains free

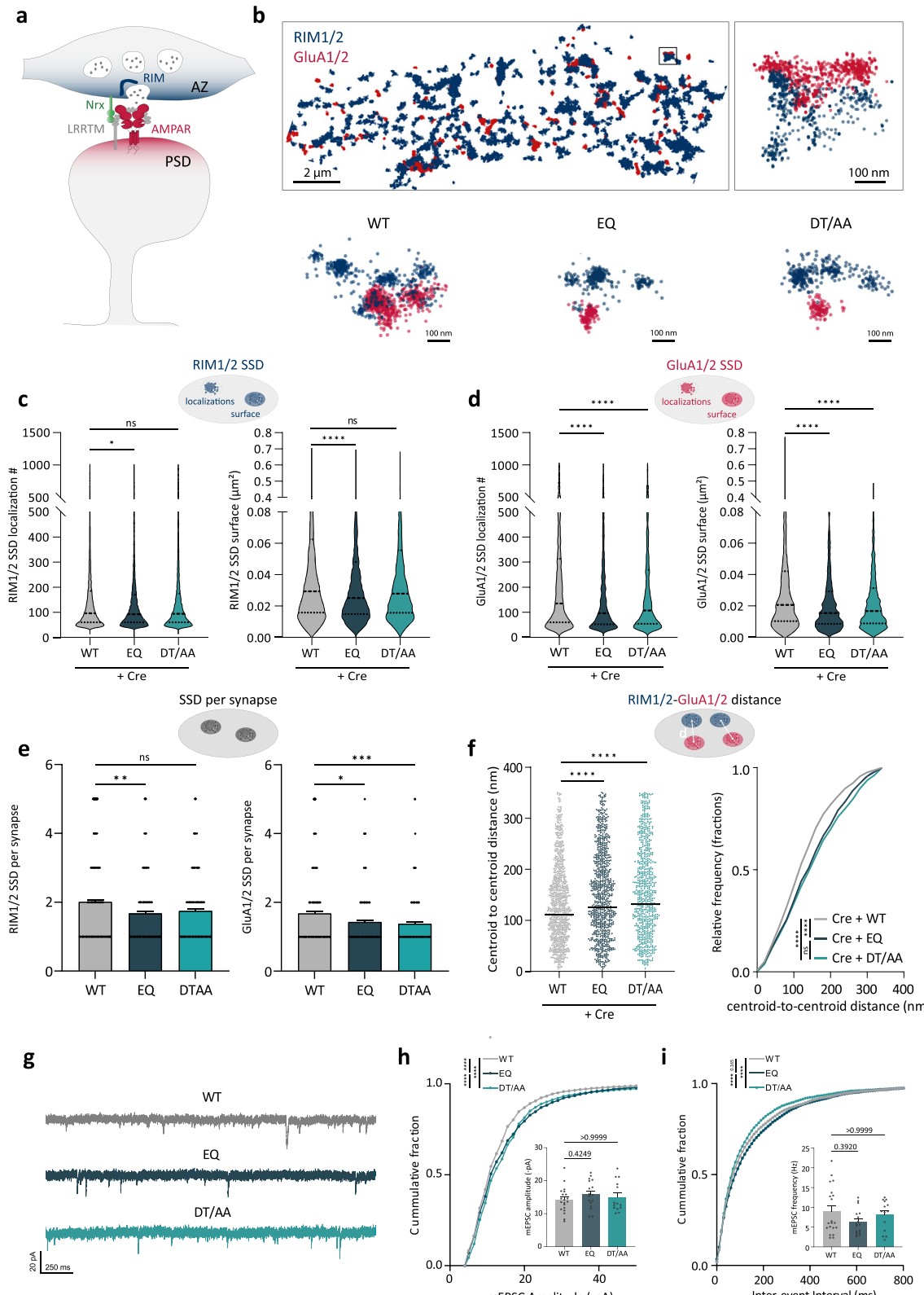

and may provide an extended binding interface important for AMPAR stabilization by direct or indirect interactions compatible with Nrxn binding. LRRTM2 and AMPARs were suggested to interact through their NTDs in heterologous cells[8,16], but DT/AA mutations did not disrupt AMPAR co-IP[16], and such interaction has never been shown in neurons so far. Additionally, LRRTM4 but not LRRTM2 was found in native AMPAR complexes, potentially reflecting the low abundance of

LRRTM2 or the difficulty of detecting the protein in these experiments[38]. Here, we detected GluA2 along with all Nrxn isoforms and PSD-95 in the proteome of LRRTM2 in hippocampal neurons using a biotinylation strategy to selectively target the surface pool of LRRTM2. These results indicate that AMPARs are present in LRRTM2 surface proteome, but further studies will be necessary to confirm these findings and determine whether these proteins directly

**Fig. 6 | The Neurexin-binding site E348 is required for the nanoscale organization of presynaptic RIM scaffolds and postsynaptic AMPA receptors.**
**a** Schematics of trans-synaptic nanocolumns, where postsynaptic AMPARs anchored by PSD-95 scaffolds are aligned in front of release sites organized by presynaptic scaffolds such as RIM. **b** Example of reconstructed images from dual-color dSTORM of endogenous RIM1/2 (blue) and GluA1/2 (red) in the control condition (WT), showing apposition between RIM and AMPAR nanoclusters. Below are representative examples of RIM1/2-GluA1/2 apposition in different conditions showing disruption of GluA1/2 nanoscale organization in the presence of EQ and DT/AA compared to WT, and selective disruption of presynaptic RIM nano-organization in EQ condition. **c** Quantifications of RIM1/2 localizations in sub-synaptic densities (SSDs) and RIM1/2 cluster surface, showing a selective decrease in EQ condition, but not in DT/AA condition. **d** Quantifications of GluA1/2 localizations in SSDs, and cluster surface, showing a decrease with both EQ and DT/AA mutants. **e** Quantifications of RIM1/2 and GluA1/2 SSD number per synapse, depicting a specific decrease of RIM1/2 SSD only with the EQ mutant. Data

presented as mean values ± SEM. **f** Quantifications of the mean distances between RIM1/2 and GluA1/2 SSD centroids and relative frequency distribution of these distances, showing the increased distance between presynaptic scaffolds and postsynaptic AMPARs in the mutant conditions. Data obtained from three independent experiments (WT: $n = 8$; EQ: $n = 10$; DT/AA: $n = 8$ cells), $*p < 0.05$, $**p < 0.01$, $***p < 0.001$, $****p < 0.0001$. Data were compared by one-way analysis of variance test, followed by post-hoc Dunn's test. **g** Representative mEPSC traces recorded from DIV15 hippocampal neurons expressing Cre-GFP, BirA[ER], and WT-, EQ-, or DT/AA- LRRTM2. **h** Cumulative graph and plot of mean mEPSC amplitudes, showing a significant shift of mEPSC amplitudes in the mutant conditions compared to WT. Data presented as mean values ± SEM. **i** cumulative graph of inter-event interval and plot of mean mEPSC frequency. Data presented as mean values ± SEM. Data acquired from three independent experiments (WT, $n = 20$; EQ, $n = 17$; DT/AA, $n = 14$) $****p < 0.0001$. Data were compared by one-way analysis of variance test, followed by post-hoc Dunn's test.

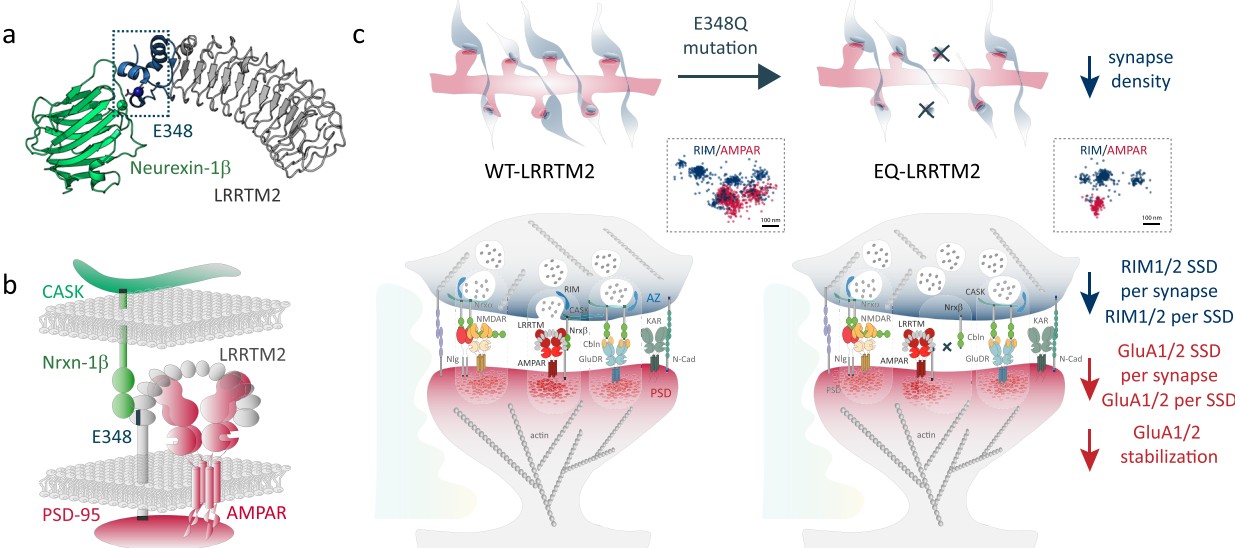

**Fig. 7 | Summary model. a** Crystal structure of LRRTM2 (gray) in complex with Neurexin-1β (green) (PDB 5Z8Y) showing the interaction site with glutamic acid E348[14] and a calcium ion (dark blue and dark green, respectively, boxed region). **b** Model of LRRTM2-containing nanocolumn showing interaction with Nrxn1β through E348, the interaction of Nrxn1β with scaffold molecule CASK, and of

LRRTM2 with PSD-95 through PDZ-binding motifs (black), and the concave interface of LRRTM2 which could stabilize GluA2-containing AMPARs at synapses. **c** Working model: disruption of LRRTM2-Nrxn1β binding upon E348Q mutation induces a loss of synapse density. At the synaptic level, LRRTM2-dependent AMPAR and RIM SSDs are lost upon EQ mutation.

interact, and what is the influence of Nrxn-binding site E348 on this interaction. Indirect interactions may be mediated by other molecular partners, such as proteoglycans, as LRRTM2 and Nrxns can bind in a heparan-sulfate-dependent manner[39]. Lastly, indirect stabilization of synaptic AMPARs may be achieved through recruitment of PSD-95 scaffolds, in a mechanism analogous to that reported for Neuroligin1[31,40]. Here, we show that CTD deletion reduces PSD-95 density, inducing a slight increase in synaptic AMPAR turnover, presumably through loss of stabilizing PSD-95 scaffolds, but these effects are mild compared to LRR deletion, which further reduces the slow pool fraction of synaptic AMPARs. These results indicate that the LRR domain is directly involved in AMPARs stabilization, whereas the CTD effect is indirect. Finally, although very hypothetical, LRRTM2 could be involved in synapse organization through phase separation. PSDs were proposed to assemble autonomously by liquid–liquid phase-separation through multivalent interactions[41,42]. In this context, LRRTM2 could serve to establish phase-separated subdomains into which AMPARs might partition[21]. Although phase-separation is facilitated by proteins that oligomerize, and it is not known at this stage whether this is the case for LRRTM2, this hypothesis is consistent with

the very potent clustering effect of LRRTM2 on PSD-95 observed here. Further assessment of the oligomerization capacities of LRRTM2 will be valuable to understand these mechanisms. To conclude, our results provide important insights into the molecular mechanisms of trans-synaptic nanocolumn organization by synaptic adhesion complexes and resolve some controversial observations. Further exploration of the individual functions and nanoscale organization of other synaptic CAMs and their interactors will contribute to a better understanding of the molecular mechanisms by which synapses co-assemble and fine-tune their architecture to maintain neuronal connectivity.

## Methods

### LRRTM2 floxed mouse line
LRRTM2[flox/flox] mouse line was outsourced to Ingenious Targeting Laboratory. The targeting vector for homologous-based recombination was generated using mouse BAC-clone RP23-325I24 (chr18, 35352957–35567755). The vector contained two loxP sites flanking exon2 and a Neomycin resistance cassette flanked by Frt sites. Recombinants were screened and confirmed by colony PCR and sequencing. Targeting vector was transfected into embryonic stem

(ES) cells for homologous recombination. ES selections were performed using G418. Positive ES cells were expanded and verified by Southern blot. Positive clones were injected into C57BL/6J blastocysts and implanted in female mice. Founder chimeras were backcrossed with C57BL/6J mice. Floxed mice were maintained on a C57BL/6J background (Charles River). All experiments were performed according to the European guide for the care and use of laboratory animals and animal care guidelines issued by the animal experimental committee of Bordeaux Universities (CE50; A5012009).

## Plasmids and viruses

AP-LRRTM2 -WT, -ΔC, -ΔECEV, -YACA were described[5,12]. AP-ΔLRR-LRRTM2 was derived from myc-ΔLRR-LRRTM2[8] replacing myc- by AP-tag (GLNDIFEAQKIEWHE). SEP-LRRTM2 was generated by changing YFP to SEP in YFP-LRRTM2 (A.M Craig, University of British Columbia). AP-EQ-LRRTM2, AP-DT/AA-LRRTM2, and SEP-LRRTM2-YACA were generated by site-directed mutagenesis. SEP-GluA1 and SEP-GluA2 were from Yukiko Goda (RIKEN CBS). Homer1c-DsRed, Homer1c-BFP, and PSD-95 GFP were described[40]. BirA$^{ER}$ was from A. Ting (Stanford). pET-IG-mSA (Addgene #80706) was described[23]. pEGFP-C1 (clontech #6084-1) and pmCherry-C1 (clontech #632524) from Clontech. pAAV-Ef1a-mCherry-IRES-Cre from D. Choquet (IINS, addgene #55632), pAAV-BirAER was described[43], pAAV-CamKII-AP-LRRTM2 and pAAV-Ef1a-GFP-IRES-Cre were generated in the laboratory. pENN.AAV.h-Syn.Cre.WPRE.hGH (addgene #105553-AAV9) was from C. Mulle (IINS). pAAV−hSyn-Cre-T2A-EGFP was generated from pENN.AAV.hSyn.-HI.eGFP-Cre.WPRE.SV40 (addgene #105540). AAVs were produced by the viral core facility of Bordeaux Neurocampus IMN, with titers around $10^{14}$ GCP/ml.

## Proteins

Nrxn1β-Fc (-SS4)[32] was purified from a conditioned medium of stable hygromycin-resistant HEK-293 cell-line[44] using HiTrap-ProteinG-HP (GE-Healthcare) to a concentration of 0.6–1.0 mg ml$^{-1}$. mSA was produced and conjugated with STAR635P, Atto 565 and Alexa 647 as described[22,23], and concentrated to 0.2 mg ml$^{-1}$ using Amicon Ultra centrifugal filters with a 10-kDa cutoff. Streptavidin-AlexaFluor405 (# S32351) and streptavidin-AlexaFluor555 (#S-21381) were from Thermo Fisher Scientific. Antibodies were mouse-α-PSD-95 (Thermo Fischer Scientific, clone 7E-1B8, 1:400), guinea-pig-α-VGluT-1 (AB5905, Merck Chemicals, 1:1000), mouse-α-GluA1/2 (Synaptic Systems, 182411, 1:100), mouse-α-gephyrin (SynSys, 147111, 1:1000), rabbit-α-RIM1/2 (SynSys, 140213, 1:500), mouse-α-synapsin-1 (Synaptic System, 106011, 1:500), sheep-α-LRRTM2 (R&D Systems, AF5589, 1:200), mouse-α-β-actin (Sigma-Aldrich #A5316), rabbit-α-streptavidin (Rockland, 100-4195), rabbit-α-GFP (Abcam Ab-290), AlexaFluor®647-Goat-α-Human-IgG (Jackson ImmunoResearch Laboratories, 109-605-098), horseradish peroxidase (HRP)-donkey anti-secondary antibody (Jackson Immunoresearch; 713-035-003), IRDye® 680RD Streptavidin (LI-COR #926-68079), IRdye-800CW goat-α-rabbit (LI-COR # 926-32211), goat-α-mouse-AlexaFluor488 (ThermoFischer Scientific, #A11001 1:800), goat-α-guinea-pig-DyLight405 (Abberior, #106-475-003, 1:800), goat-α-mouse-IgG2a-AlexaFluor647 (Thermo Fisher Scientific #A21241), goat-α-rabbit-AlexaFluor647 (Thermo Fisher Scientific #A21244) and CF®597R-goat-α-rabbit (biotium #20797).

## Neuron cultures

Banker cultures of primary hippocampal neurons from P0 LRRTM2$^{Flox/Flox}$ mice, regardless of sex were cultured as described[45]. Neurons were plated at 500,000 cells per 60-mm dish, on pre-coated 18-mm 1.5H coverslips (Marienfeld-Superior, 0117580) in Neurobasal™-A (NB-A) medium (Thermo Fisher Scientific), supplemented with NeuroCult™ SM1 (STEMCELL), 2 mM glutamax (Gibco Thermofisher, #35050061) and 10% Horse Serum for 30 min, then serum was removed and coverslips flipped onto 60-mm dishes containing astrocyte monolayer and

cultured 2 weeks at 37 °C and 5% $CO_2$. Cytosine arabinoside (3.4 mM) was added at DIV3 to control glial growth. For biochemistry, cells were plated at 500,000 cells/well in pre-coated six-well plates. After 30 min, the medium was replaced with supplemented Neurobasal™-A-3% Horse Serum, and 72 h later, the medium was partially replaced by supplemented Neurobasal™ without serum. Cytosine arabinoside (3.4 mM) was added at DIV3. Neurons were transfected at DIV 7 using calcium phosphate. 1.5–1.8 µg of plasmid DNA was mixed with the following solutions: TE (1 M Tris–HCl pH 7.5, 250 mM EDTA), CaCl₂ (2.5 M $CaCl_2$ in 10 mM HEPES−pH 7.2) and 2xHEPES-buffered saline (274 mM NaCl, 10 mM KCl, 1.4 mM Na₂HPO4, 12 mM glucose, 42 mM HEPES−pH 7.2). Coverslips were transferred to 12-well plates containing cultured medium, 2 mM kynurenic acid, and DNA mix. After 30-min incubation at 37 °C, cells were washed with a medium containing 2 mM kynurenic acid and returned to their original dish. Organotypic hippocampal slice cultures were prepared from postnatal day 5 (P5) LRRTM2$^{Flox/Flox}$ mice as previously described[46]. Hippocampi were dissected and coronal slices (350 µm) were cut using a tissue chopper (McIlwain) and incubated at 35 °C with the serum-containing medium on Millicell culture inserts (CM, Millipore). The medium was replaced every 2–3 days.

## Heterologous cells, Nrxn clustering, SEP-GluA2 cross-linking, and co-cultures

COS-7 cells were plated at a density of 50,000 cells/well in 12-well plates containing sterile glass coverslips, cultured in DMEM (GIBCO/BRL) supplemented with 10% fetal bovine serum (Eurobio), 100 units ml$^{-1}$ penicillin, 100 mg ml$^{-1}$ streptomycin and biotin 10 µM, at 37 °C with 5% of $CO_2$. 3–6 h after plating, transfections were done with X-treme GENE™ Transfection Reagent (Roche). 1 µg total DNA (0.4 µg AP-LRRTM2-WT or mutants + 0.4 µg BirA$^{ER}$ ± 0.2 µg PSD-95-GFP or EGFP) was mixed with 2 µl X-treme gene reagent in 100 µl PBS, and incubated at room temperature (RT) 30 min. 30 µl was added to cells per 1 mL volume. After 24 h, cells were incubated with 100 mM STAR635P- or ATTO 565-mSA in Tyrode's solution (in mM: 15 D-glucose, 108 NaCl, 5 KCl, 2 MgCl₂, 2 CaCl₂, and 25 HEPES, pH 7.4) containing 0.1% biotin-free BSA (Carl Roth) 10 min at RT before live imaging. For the Nrxn-binding assay, prior to mSA-labeling, cells were incubated for 20 min with Nrxn solution (2 µg Nrxn-1β-Fc + 1.3 µg AlexaFluor647-α-humanFc). For SEP-GluA2 cross-linking, COS-7 cells were incubated for 30 min in Tyrode's solution containing 0.2 µg rabbit-anti-GFP + 0.1 µg goat-anti-rabbit-AF647 + 0.1 µg streptavidin-AF555, and imaged live. For co-cultures, transfected COS-7 cells were detached after 24 h, resuspended in NB-A-SM1 medium, and plated directly onto DIV7-9 hippocampal neurons. Cells were left to adhere for 30 min and coverslips were flipped back onto astrocyte monolayers and cultured 24–48 h before fixation and immunolabeling.

## Biochemistry

Primary hippocampal neurons and organotypic slices were infected at DIV6 for 6 h with AAV-hSyn-Cre-T2A-EGFP at 30,000 MOI. Samples were lysed 6–7 days post-infection in lysis buffer (HEPES 500 mM, NaCl 100 mM, Glycerol 1x, dichloro-diphenyl-trichloroethane 14 mM, 1x protease inhibitor cocktail (Sigma-Aldrich, #P2714) for 2 h at 4 °C on a rotating device. Lysates were centrifuged at 13,000×$g$ for 15 min at 4 °C. Protein concentrations were quantified using Direct DetectTM Infrared Spectrophotometer (Merck-Millipore), and protein amounts were adjusted for loading. Samples were warmed for 10 min at 70 °C, before loading on 4–20% SDS−PAGE gel (Biorad). Proteins were transferred onto nitrocellulose membrane and incubated in blocking solution (LI-COR) for 1 h at RT, before incubation with primary anti-LRRTM2 overnight at 4 °C and HRP-donkey anti-secondary antibody. β-actin was used as the loading control. Target proteins were detected by chemiluminescence using Clarity MAX Western ECL Substrate (BioRad) on the ChemiDoc Touch system (Bio-Rad). Average intensity values were calculated using Image Lab 5.0 software (Bio-Rad).

LRRTM2 signal intensity was normalized to β-actin. For co-immunoprecipitation experiments, hippocampal neurons were infected at DIV6 for 6 h with AAV-hSyn-Cre-T2A-EGFP, AAV-BIRAER, and AAV-CamKII-AP-LRRTM2 (30,000 MOI). At DIV21, cells were rinsed in ACSF-0.1% BSA w/o biotin (Carlroth) and incubated with streptavidin (100 μM) 40 min at 4 °C, washed, and quenched 5 min with biotin (50 μM) before lysis (50 mM Tris−HCl pH 7.4, 150 mM NaCl, 2 mM MgCl$_2$, 2 mM CaCl$_2$, 1% TritonX-100, 1x protease inhibitor cocktail (Sigma-Aldrich, #P2714)) 15 min on ice. Cells were scraped, and lysis continued on a rotating wheel at 4 °C for 40 min, before centrifugation at 8000×$g$ 15 min at 4 °C. Protein concentrations were quantified using Direct DetectTM Infrared Spectrophotometer. 10 μg rabbit-anti-streptavidin per 600 μg protein was incubated for 2 h at 4 °C. 50 μg sheep-anti-rabbit magnetic beads (Thermo Fisher Scientific, 11203D) were added to the lysate and incubated overnight at 4 °C. Beads were immobilized on a magnetic holder, washed and eluted in 50 μL loading blue 2X (Sigma Aldrich, S3401-10VL). Pull-downs were performed on COS-7 cells transfected 48 h with X-treme GENE™ (Roche) with DNA ratio 1:0.3:0.2 of SEP-GluA2:WT- or ΔLRR-AP-LRRTM2:BirA$^{ER}$. Cell were lysed in 50 mM Tris−HCl pH 7.4, 150 mM NaCl, 1% TritonX-100, 1x protease inhibitor mix (Sigma-Aldrich, #P2714), incubated 40 min at 4 °C, and centrifuged at 8000×$g$ 15 min. 40 μl Dynabeads™ M-280 Streptavidin (ThermoFisher #11205D) per sample were incubated with protein lysis 1 h30 at 4 °C on ThermoMixer™ (ThermoFisher) at 1400 rpm. Samples were eluted in 2x Laemmeli (Bio-rad), warmed at 95 °C 5 min, and loaded on 4−20% SDS−PAGE gel (Biorad), before transfer onto nitrocellulose membrane and incubation in blocking solution (LI-COR), followed by primary anti-GFP and IRDye 800 secondary antibody solutions, and IRDye 680 streptavidin. Protein detection was performed with the LI-COR Odyssey FC system (LI-COR).

## nLC−MS/MS analysis and label-free quantitative data analysis

Protein samples were digested with Trypsin Gold (Promega) and solubilized in 0.1% HCOOH. Peptides were analyzed on Ultimate 3000 nanoLC system (Dionex, Amsterdam, The Netherlands) coupled to Electrospray Orbitrap Fusion™ Lumos™ Tribrid™ Mass Spectrometer (ThermoFisher Scientific), after loading onto 300-μm-inner diameter x 5-mm C18 PepMapTM trap column (LC Packings) at a flow rate of 10 μL/min and eluted from the trap column onto an analytical 75-mm i.d. × 50-cm C18 Pep-Map column (LC Packings) with a 5−27.5% linear gradient of solvent B in 105 min (solvent A: 0.1% formic acid, solvent B: 0.1% formic acid in 80% ACN) followed by a 10 min gradient from 27.5% to 40% solvent B. The separation flow rate was 300 nL/min. The mass spectrometer operated in positive ion mode at a 2-kV needle voltage. Data were acquired using Xcalibur software in a data-dependent mode. MS scans ($m/z$ 375−1500) were recorded in the Orbitrap at a resolution of $R = 120,000$ (@ $m/z$ 200) top speed fragmentation in HCD mode was performed over a 3 s cycle. MS/MS scans were collected in the Orbitrap with a resolution of 30,000, a normalized HCD collision energy of 30% and an isolation width of 1.6 $m/z$. Data were searched by SEQUEST through Proteome Discoverer 2.5 (Thermo Fisher Scientific Inc.) against Mus musculus uniprot database (17,050 entries in v2021-01). Search parameters were: mass accuracy of monoisotopic peptide precursor and peptide fragments was set to 10 ppm and 0.02 Da, respectively. Only b- and y-ions were considered. Sequest HT was used as the search algorithm: Oxidation of methionines (+16 Da), methionine loss (−131 Da), methionine loss with acetylation (−89 Da), protein N-terminal acetylation (+42 Da) were considered as variable modifications while carbamidomethylation of cysteines (+57 Da) was considered as fixed modification. Two missed trypsin cleavages were allowed. Peptide validation was performed using Percolator algorithm[47], and only "high confidence" peptides were retained, corresponding to a 1% false positive rate at the peptide level. Peaks were detected and integrated using the Minora algorithm embedded in Proteome Discoverer.

## RT-qPCR

Banker cultures were infected at DIV6 using different MOIs of pEN-N.AAV.hSyn.Cre.WPRE.hGH for RT-qPCR as indicated in figure legend for 6 h. Cultures were lysed at DIV14 using QIAzol Lysis Reagent (Qiagen), and RNA was isolated with the Direct-Zol RNA microprep (Zymo Research) according to the manufacturer's instructions. cDNA was synthesized using the Maxima First Strand cDNA synthesis kit (Thermo Fischer Scientific). At least two neuronal cultures were analyzed per condition, and triplicate qPCR reactions were made for each sample. Transcript-specific primers were used at 2 μM and cDNA at 5 ng in a final volume of 10 μL. The LightCycler 480 ONEGreen® Fast qPCR Premix kit (Ozyme) was used according to the manufacturer's instructions. The Ct value for each gene was normalized against that of Succinate Dehydrogenase Complex Flavoprotein Subunit A (SDHA). The relative level of expression was calculated using the comparative method (2-ΔΔCt)[48]. The following primers were used: LRRTM2 Forward: 5′CCAATTTCCGAGGCAAACC 3′ Reverse: 5′CACACTCAAAGTC TTTCCCTG 3′ and SDHA Forward: 5′ TGCGGAAGCACGGAAGGAGT 3′ Reverse: 5′ CTTCTGCTGGCCCTCGATGG 3′.

## Electrophysiology

Electrophysiological recordings were carried out at RT on primary hippocampal neurons from LRRTM2$^{Flox-Flox}$ mice transfected with an empty vector (control), a plasmid encoding Cre-mCherry reporter alone (cKO) or with WT-, EQ-, or DT/AA-LRRTM2. Neurons were observed with an upright microscope (Nikon Eclipse FN1) equipped with a motorized 2D stage and micromanipulators (Scientifica). Whole-cell patch-clamp was performed using micropipettes pulled from borosilicate glass capillaries using a micropipette puller (Narishige). Pipettes had a resistance in the range of 5−6 MΩ. The recording chamber was continuously perfused with aCSF containing (in mM): 130 NaCl, 2.5 KCl, 2.2 CaCl$_2$, 1.5 MgCl$_2$, 10 D-glucose, 10 HEPES, and 0.02 bicuculline (pH 7.35, 300 mOsm). The internal solution contained (in mM): 135 Cs-MeSO$_4$, 8 CsCl, 10 HEPES, 0.3 EGTA, 4 MgATP, 0.3 NaGTP, and 5 QX-314. Salts were from Sigma-Aldrich and drugs from Tocris. Neurons were voltage-clamped at a membrane potential of −70 mV, and AMPAR-mediated mEPSCs were recorded in the presence of 0.5 μM TTX using Clampex (Axon Instruments). The series resistance Rs was left uncompensated. Recordings with Rs higher than 30 MΩ and changes >20% were discarded. mEPSCs were detected and analyzed using MiniAnalysis (Synaptosoft).

## Immunocytochemistry and epifluorescence microscopy

Neurons were live-labeled 10 min at RT with STAR635P-conjugated-mSA and/or anti-GluA1/2 in ACSF (in mM: 130 NaCl, 2.5 KCl, 2.2 CaCl$_2$, 1.5 MgCl$_2$, 10 D-glucose, 10 HEPES), fixed 10 min in 4% paraformaldehyde−20% sucrose, quenched in NH$_4$Cl 50 mM 10 min, and permeabilized 7 min with 0.1% Triton X-100. Non-specific binding was blocked using 1% biotin-free BSA (carlroth) for 45 min. Neurons were immunostained for 1 h with primary antibodies followed by 1-h incubation with secondary antibodies. Coverslips were mounted in Fluoromount™ (Merck). Labeling of endogenous LRRTM2 was achieved using Glyoxal fixation as previously described[49]. Neurons were fixed for 1 h at RT with 3% Glyoxal (128465, Merk) and 0.8% acetic acid, adjusted to pH 4.5. After fixation, neurons were rinsed in PBS-T containing 0.1% Triton X-100 (Merk), incubated in NH$_4$Cl 50 mM for 10 min, permeabilized and blocked in PBS-T solution containing 2% BSA 30 min, and incubated with primary antibody anti-LRRTM2 in PBS-T 1 h followed by secondary antibody incubation 1 h. Immunostained neurons were visualized using an inverted epifluorescence microscope (Nikon Eclipse TiE) with a ×60/1.40 NA objective and filter sets for BFP (excitation: FF01-379/34; dichroic: FF-409Di03; emission: FF01-440/40); EGFP (Excitation: FF01-472/30; Dichroic: FF-495DiO2; Emission: FF01-525/30); mCherry (Excitation: FF01-543/22; Dichroic: FF562DiO2; Emission: FF01-593/40); and Alexa647 (Excitation: FF02-628/40;

Dichroic: FF-660Di02; Emission: FF01-692/40) (SemRock). Images were acquired with Prime 95BTM sCMOS (Phototometrics®) under Metamorph® (Molecular Devices). Transfected cells were identified with GFP or mCherry reporter. For replacement conditions, only Cre-mCherry- or Cre-GFP- positive cells and AP-LRRTM2 were considered. To measure protein levels, a region was drawn around reporter fluorescence, and transferred to the signal after background subtraction. Intensities were normalized to the control condition. To calculate cluster density, regions around reporter-positive dendrites were created, and the number of clusters per unit length was measured using MetaMorph. For synapse density, pre- and postsynaptic masks were segmented and dilated after background subtraction and merged to detect intersected areas using MetaMorph. Representative dendritic fragments shown in the figures are chosen from primary or secondary dendritic branches.

## Single particle tracking

Cells were mounted in Tyrode's solution containing 0.1% biotin-free BSA in an observation chamber (Life Imaging Services, Basel) on an inverted microscope (Nikon Ti-E Eclipse) equipped with an Evolve EMCCD camera (Roper Scientific, France), a thermostatic box (Life Imaging Services) at 37 °C, an APO total internal reflection fluorescence (TIRF) ×100/1.49 NA oil objective, and a four-color laser bench (405; 488; 561; 100 mW, 642 nm, 1 W; Roper Scientific) connected through an optical fiber to the TIRF arm of the microscope. Laser powers were controlled through acousto-optical tunable filters driven by Metamorph. GFP-expressing cells were detected using a mercury lamp (Nikon Xcite) and appropriate filter sets described above (SemRock). To track biotinylated AP-LRRTM2, 1 nM STAR635P-mSA was used. Samples were imaged by oblique laser illumination, allowing excitation of individual STAR635P-conjugated ligands bound to the cell surface[24]. Stacks of 2000 consecutive frames were obtained from each cell, with an integration time of 20 ms. Single-molecule information from image stacks was extracted using a custom program in Metamorph based on wavelet segmentation described earlier[50,51]. The diffusion coefficient was calculated for each trajectory from linear fits of the first four points of the mean square displacement (MSD) function versus time using a custom routine written in Matlab (Math-Works). We analyzed trajectories lasting at least 10 points (≥200 ms). Trajectories with a diffusion coefficient $D < 0.005$ µm² s⁻¹ corresponding to molecules exploring an area inferior to that defined by image spatial resolution $\sim (0.04$ µm$)^2$ during the time used to fit the initial slope of the MSD[52] (4 points, 80 ms): $D_{threshold} = (0.04$ µm$)^2/(4 \times 4 \times 0.02$ s$) \sim 0.005$ µm² s⁻¹ were considered immobile. The remaining MSDs were fitted with the following equation:

$$MSD(t) = 4/3 r^2 \text{conf}(1 - e - t/\tau)$$

where $r_{conf}$ is the measured confinement radius, $\tau$ the time constant $\tau = (\frac{r^2_{conf}}{3D_{conf}})$. Confined and free-diffusing were defined as trajectories with a time constant τ respectively inferior or superior to half the time interval used to compute the MSD (100 ms), as previously described[53]. Diffusion coefficients of diffusing molecules, $D_{diff}$, were computed from linear fits of the first four points of the MSDs for tracks sorted as free-diffusive. Synapses were identified by thresholding the Homer1c-GFP signal. A trajectory was considered synaptic when spending >50% of its duration inside Homer1c-GFP-positive areas. The mean time spent at synapses was calculated as the number of frames a molecule was detected inside a synaptic area multiplied by the integration time (20 ms) for each trajectory crossing a synapse, using Matlab.

## FRAP

Neurons were mounted in an imaging chamber perfused with ACSF and observed under the same set-up described above. The laser bench has a second optical fiber output connected to an illumination device containing galvanometric scanning mirrors (ILAS, Roper Instrument) controlled by MetaMorph, enabling precise photobleaching of regions of interest. After acquiring a 10-s baseline at 1 Hz, photobleaching of 5 spines and 2 shaft regions was achieved at high laser power. Fluorescence recovery was recorded immediately after the bleach for 10−15 min at 0.5−1 Hz frame rate. For SEP-GluA1/2 FRAP, the presence of AP-LRRTM2 was verified at the end of the acquisitions using AlexaFluor647-streptavidin. The fluorescence intensity of all regions was background-subtracted and bleach-corrected. Intensities for each frame before photobleaching (10 frames/region) were normalized to their mean intensity value and to 1, and the intensity of the first frame after the photobleaching was normalized to 0. Regions with negative values after normalization or bleaching depth <50% were excluded from the analysis. Each FRAP curve was fitted as previously described[54] using the following equation in GraphPad Prism:

$$F(t) = F(0) + p\left(1 - e^{-\frac{t}{\tau f}}\right) + q\left(1 - e^{-\frac{t}{\tau s}}\right)$$

where $F(0)$ is the fluorescence intensity at time 0, $p$ and $q$ are the fractions of fast and slow pools, and τf and τs are the recovery time constants for the fast and slow pools. The slow pool fractions were extracted from each fitted recovery curve in GraphPad Prism and pooled per condition. In GluA FRAP experiments, the initial surface intensity of SEP-GluA signal in regions of interest before photobleaching was calculated by subtracting the background from the average intensity in the first frame. To assess the overall surface intensity of proteins in dendrites, a region was drawn around the Cre-mCherry signal and transferred to SEP-GluA1 images. To study SEP-LRRTM2 exocytosis in COS-7 cells, the whole field of view was bleached, and fluorescence recovery from intracellular pools (non-fluorescent intracellular SEP-proteins are protected from the bleach) was monitored at 0.5 Hz for 120 s. Maximum projections of acquired stacks were segmented using Metamorph and exocytic events were analyzed, and their decay fitted with Matlab.

## pH protocol

Neurons were perfused with 37 °C (1) ACSF pH 7.4 (baseline 1), (2) ACSF pH5.5 to quench surface SEP-tagged proteins, (3) ACSF pH 7.4 (baseline 2), (4) 50 mM NH₄Cl to reveal all fluorescent signal, and (5) ACSF pH 7.4 (baseline 3) and imaged at 0.1 Hz. Intracellular pools were extracted as the mean fluorescence under NH₄Cl (total pool) minus the mean of the three baseline signals (surface pool). Objects detected at Homer1c-positive spines were considered in spines, whereas the rest were classified as dendritic (shaft). The proportion of intracellular pools at spines was calculated as the ratio between intracellular pools inside Homer1c-positive spines divided by the total intracellular fluorescence. Surface SEP-LRRTM2 intensity was measured by drawing a region around Cre-mCherry and transferring it to SEP-LRRTM2 images.

## dSTORM

Cells were live-labeled with mouse α-GluA1/2 in ACSF for 10 min at RT and fixed for 10 min in 4% paraform-aldehyde−20% sucrose−0.2% glutaraldehyde. After blocking in PBS−BSA 1%, cells were incubated with α-mouse-IgG2a-AlexaFluor647 and streptavidin−AlexaFluor405. Cells were then permeabilized with 0.1% Triton X-100 and labeled with α-RIM1/2 followed by CF®597R-goat-α-rabbit. Coverslips were mounted in an oxygen-scavenging buffer (Tris−HCl buffer pH 7.5 containing 10% glycerol, 10% glucose, 0.5 mg/mL glucose oxidase (Sigma), 40 mg/mL catalase (Sigma C100-0.1% w/v), and 50 mM β-mercaptoethylamine (MEA) (Sigma M6500))[55] and sealed. The chamber was placed on an inverted microscope (Nikon Ti-E Eclipse) equipped with APO TIRF ×100/1.49 NA oil objective, an ORCA-Flash4.0 LT3 Digital sCMOS camera (Hamamatsu), a 500 mW 640 nm laser (Oxxius), a 1200 mW 405 nm laser (Oxxius) and a 100 mW 560 nm laser

(Roper scientific). Dual-color imaging of Alexa647 and CF597R[56] was achieved by alternating laser excitation frame by frame using an AOTF at a frequency of 50 Hz. 100-nm nano-diamonds (Adamas Nanotechnologies) were used to register long-term acquisitions and correct for lateral drift. A series of 40,000 frames were acquired using Metamorph, and processed for single-molecule localization, drift correction, data visualization, and cluster analysis using Abbelight NEO software (Abbelight) DBSCAN with the following parameters, RIM1/2: radius = 80 nm, minPts = 40; GluA1/2: radius = 70 nm, minPts = 30. To identify synapses, low-resolution images of RIM1/2 and GluA1/2 taken before STORM were segmented, dilated, and overlaid. Synapse region masks were created as the intersection of these signals. RIM1/2 and GluA1/2 SSDs per synapse were counted within these regions using Matlab. Centroid-to-centroid distances between RIM1/2 and GluA1/2 SSDs were computed for each DBSCAN-segmented GluA1/2 SSD to the nearest RIM1/2 SSD within a 350 nm radius, using Matlab.

## Statistics

Statistical values are given as mean ± s.e.m., unless otherwise stated. Statistical significance was calculated using GraphPad Prism. All data sets comparing two conditions were tested with the non-parametric Mann–Whitney test. Data sets containing more conditions were compared by one-way analysis of variance test, followed by post hoc Dunn's test, and by two-way analysis of variance test, followed by Tukey's multiple comparison test for datasets comparing different mobility fractions across conditions (Fig. 2). Sample size was based on two to three distinct cultures per condition and at least 2–10 cells per experiment. Randomization of samples was performed for all experiments. Analysis was performed blind.

## Reporting summary

Further information on research design is available in the Nature Portfolio Reporting Summary linked to this article.

## Data availability

Data supporting the findings of this manuscript are available from the source data file and corresponding author. Source data are provided with this paper. The mass spectrometry proteomics data have been deposited to the ProteomeXchange Consortium via the PRIDE[57] partner repository with the dataset identifier PXD054623. Source data are provided with this paper.

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

## Acknowledgements

This research was financed by grants from the Agence Nationale de la Recherche nanoPROBE, ANR-19-CE11-0025, SyntheSyn, ANR-17-CE16-0028-01, Fondation pour la Recherche Médicale (DEQ20160334916), and Labex BRAIN. We thank the IINS cell culture facility (E. Verdier, C. Desquines, and A. Caralp for cell cultures); the animal facility of the University of Bordeaux (S. Pavelot, P. Costet, and C. Martin); the Bordeaux Neurocampus biochemistry Platform at the Bordeaux University funded by the LABEX BRAIN (ANR-10-LABX-43) for Western blot analysis and protein quantifications; the AAV production platform of Bordeaux Neurocampus (N. Dutheil); R. Sterling and A. Castets for technical support. We thank A.M Craig, Y. Goda, D. Choquet, and C. Mulle for the generous gift of plasmids and viruses, V. Studer for the expertise and help with Matlab analysis.

## Author contributions

K.L. and I.C. designed research, performed experiments and analysis, and wrote the article. M. Lubas and C.J. performed immunocytochemistry experiments and analysis, V.V. performed electrophysiology experiments and analysis, J.C. performed biochemistry experiments, C.D. performed co-culture assays and analysis, M.M. and B.T. created plasmids and produced proteins. S.C. provided expertise and performed MS analysis. A.F. provided expertise and designed sequences for qRT-PCR. M.S. provided expertise in protein chemistry and produced monomeric streptavidin preparations. M. Letellier provided expertise and performed electrophysiology experiments. J.D.W. designed and created the cKO mouse line and provided scientific expertise. O.T. provided scientific expertise and financial support. All authors discussed the results and manuscript.

## Competing interests

The authors declare no competing interest.
