## [Peer Review File · Nature Communications]

LRRTM2 controls presynapse nano-organization and AMPA receptor sub-positioning through Neurexin-binding interfaceREVIEWER COMMENTS

Reviewer #1:

The authors report a detailed characterization of the synaptic functions of LRRTM2, where they dissect its surface expression, synaptic clustering, and membrane diffusion, determining that these parameters are controlled by different motifs in LRRTM2 C-terminal domain. In contrast, the N-terminal domain appears to control synapse nano-clustering, AMPARs sub-positioning and stabilization, through Neurexin. This work is important because LRRTM2 and the neurexins appear to be central organizers and modulators of excitatory synapses, with implications in CNS development and disease.

While some of the experiments are by necessity confirmatory of previously published material, this work shed more light on many of the details about the functional role of LRRTM2 and NRXN and their interaction. I want to commend the authors for the impressive state of the art tools and techniques they employ, the quality of the data, followed by rigorous analyses and interpretation. Overall, I don't have major issues with this work other than a few minor points detailed below.

For the details of figure 5, I suggest to add the PDB ID for the NRXN-LRRTM2 crystal structure so that the interested readers can quickly search and look at the structure themselves. It would be also useful to replace the LRRTM2 structure of panel 5a with the model that includes the β NRXN structure, thus including the residues important for the interaction.

The bottom panel in figure 5C is covered by a white stain of uncertain origin. Please clarify what this is and correct it.

Please make it easier for the reader to understand the curves of these graphs: most of them have purple-ish and green-ish curves that are difficult to distinguish, especially in a small figure/print and one has to zoom in significantly to tell which is which (or my monitor is terrible).

This very recent paper may be worth citing here:

<https://www.ncbi.nlm.nih.gov/pmc/articles/PMC10404257/> describes how NRXN3 also takes part of subsynaptic organization from the opposite side of LRRTM2

Reviewer #2:

Recent evidence strongly supports an important role of trans-synaptic nano-columns in synaptic specification. Leucine-Rich Repeat Transmembrane protein LRRTM2, which is a major postsynaptic cell adhesion molecule at excitatory synapses, has been proposed to tether presynaptic molecular release machinery and AMPAR postsynaptic nanodomains via the interaction with presynaptic adhesion molecule Neurexin. Using advanced imaging techniques and LRRTM2 cKO-mediated replacement of endogenous LRRTM2 with their mutants, this study by Liouta et al revealed that the intracellular domain was necessary for the synaptic stabilization of LRRTM2 and AMPARs, while the extracellular domain regulated the presynaptic release machinery organization, AMPAR subsynaptic position, and AMPAR turnover via the Neurexin-binding interface at mature excitatory synapses in cultured hippocampal neurons. The current manuscript is well written, and contains some interesting results. However, the conclusion sounds like a follow-up study to validate the previous cellular or structural biological knowledge on LRRTM2. More detailed mechanisms underlying LRRTM2-mediated stabilization of synaptic AMPARs via its direct or indirect interactions with AMPARs should improve the manuscript significantly. I also found the following points to be improved.

#1 Some imaging data are not convincing. In Figure 1e, YACA, but not dECEV, mutation seems to decrease the number of AP-LRRTM2 and PSD95, compared with the wild type.

#2 The authors should provide scattered plots for individual data points in Figures 2d, 2h, 6d, and 6e.

#3 The current result is based on the replacement of endogenous LRRTM2 with exogenous one, which can potentially cause higher expression levels of LRRTM2 than its endogenous levels. The authors should present a comparison of the expression levels between endogenous and exogenous LRRTM2.

#4 The authors should demonstrate the criteria to choose dendritic fragments. The morphology of dendritic branches presented in the current figures seems to differ branch by branch.

#5 The amplitude of mEPSCs in LRRTM2 cKO hippocampal neurons will be helpful to support the authors' imaging result of synaptic GluA expressions, and also compare the synaptic phenotype with previous studies.

#6 How did the authors calculate the proportion of the intracellular pools of LRRTM2 at dendritic spines and shafts? Although this must depend on the area they analyzed, no explanation was provided.

#7 What is the molecular mechanism underlying the difference in presynaptic nano-scale organizations of RIM1/2 clusters between EQ and DAT/AA mutations?

#8 It is unclear how the authors identify synaptic areas containing both RIM1/2 and AMPAR clusters in dSTORM imaging.

#9 The authors should explain which they immunolabeled cell-surface or total populations of GluAs in dSTORM imaging in the result section.

Reviewer #3:

Liouta et al used a conditional knockout mouse model of LRRTM2 and a variety of imaging approaches to assess the role of LRRTM2 in glutamatergic synaptogenesis and receptor stabilization at synapse. The authors first characterized the role of LRRTM2 in synapse development using cKO as well as the mechanism underlying synaptic anchoring and trafficking dynamics of the molecule itself. They then examined the impacts of two mutation sites in LRRTM2 on its interaction with presynaptic Neurexins. Finally they characterized how these mutations modulate the nanoscale organization of synaptic proteins including postsynaptic AMPARs and presynaptic RIM. The experiments in the paper are technically sound and clearly presented, and provide some very interesting and important phenomenon along each lines mentioned above. However, I find many of these findings are incremental, and the whole study is premature and short at a central conceptual advance.

1. The team's previous work as well as others (e.g. Linhoff 2009, Minatohara, 2015) have demonstrated that the YxxC motif at the C-terminal domain of LRRTM2 is critical for its binding with PSD95, synapse positioning and surface diffusion. The data in figures 1-3 in the current study have only verified this in a cKO system in a technically elegant way. The most novel observation in this part is that the YxxC motif regulates the exocytosis of LRRTM2-containing vesicles but the authors did not pursue further on this.

2. The role of LRRTM2 in stabilizing AMPARs at synapses is well established by multiple studies, e.g. Soler-Llavina 2013, Aoto 2013, Bhourri 2018, and Ramsey 2021. The FRAP results in figures 4 and S5 are well predicted. Moreover, considering the CTD is key for synaptic targeting of LRRTM2, the data from neurons expressing LRRTM2 without CTD add no meaningful information on the modulation of AMPAR stabilization.

3. For the FRAP results, the recovery time constant and extent should be quantified and discussed. Also, with the small differences between groups, another control experiment in neurons with intact endogenous LRRTM2 should be performed.

4. To characterize the effects of the two mutants on Nrx binding, a simple binding assay with purified Nrx fragment is not convincing. A synaptogenesis test with cocultures and a quantification of synapse density in mutant-expressing neurons would provide much more information.

5. LRRTM2 has been proposed to stabilize AMPARs through direct binding and/or indirect scaffolds-mediated interactions. The authors got around this key question and measured how the mutations related to Nrx binding affects AMPAR stabilization. However, without a solid conclusion on how the mutants impact the LRRTM2-Nrx interaction and the synapse formation, these results lead to no deeper understanding of the mechanism underlying receptor dynamics.

6. The authors quantified many detailed changes in nanoscale organizations in figure 6. What do these structural changes mean for synaptic functions?

7. What are the functional impacts of these LRRTM2 mutants?

8. What is the expression level of the AP-tagged LRRTM2 variations compared with the endogenous protein?

9. The mixture use of different abbreviations for the same mutation is disturbing, e.g. EQ vs E348Q, DT/AA vs D259/T261, YACA vs Y500A/C503A.

Point-by-point answer to the reviewer's comments

- Reviewer #1 (Remarks to the Author):

The authors report a detailed characterization of the synaptic functions of LRRTM2, where they dissect its surface expression, synaptic clustering, and membrane diffusion, determining that these parameters are controlled by different motifs in LRRTM2 C-terminal domain. In contrast, the N-terminal domain appears to control synapse nano-clustering, AMPARs sub-positioning and stabilization, through Neurexin. This work is important because LRRTM2 and the neurexins appear to be central organizers and modulators of excitatory synapses, with implications in CNS development and disease.

While some of the experiments are by necessity confirmatory of previously published material, this work shed more light on many of the details about the functional role of LRRTM2 and NRXN and their interaction. I want to commend the authors for the impressive state of the art tools and techniques they employ, the quality of the data, followed by rigorous analyses and interpretation. Overall, I don't have major issues with this work other than a few minor points detailed below.

We thank the reviewer for positive appreciation of our work, for reviewing the present paper, and suggesting improvements to the present manuscript. We have followed all his/her suggestions. Changes are highlighted in blue in the revised manuscript version.

For the details of figure 5, I suggest to add the PDB ID for the NRXN-LRRTM2 crystal structure so that the interested readers can quickly search and look at the structure themselves. It would be also useful to replace the LRRTM2 structure of panel 5a with the model that includes the β NRXN structure, thus including the residues important for the interaction.

- We thank the reviewer for this insightful comment. We have included the PDB ID of the NRXN-LRRTM2 structure in figure 5 legend and modified panel 5a to include β NRXN structure, and highlighted the residues involved in the interaction according to the reviewer's suggestions.

The bottom panel in figure 5C is covered by a white stain of uncertain origin. Please clarify what this is and correct it.

- We removed the misleading white stain in panel 5c which was a cell expressing high levels of EGFP leading to image saturation.

Please make it easier for the reader to understand the curves of these graphs: most of them have purple-ish and green-ish curves that are difficult to distinguish, especially in a small figure/print and one has to zoom in significantly to tell which is which (or my monitor is terrible).

- We thank the reviewer for his/her help to improve figure readability. We changed colors and color tones accordingly, making it easier to understand figures and graphs.

This very recent paper may be worth citing here: <https://www.ncbi.nlm.nih.gov/pmc/articles/PMC10404257/> describes how NRXN3 also takes part of subsynaptic organization from the opposite side of LRRTM2

- We have now cited and discussed this recent paper in the discussion section of the revised manuscript (p.9).

- Reviewer #2 (Remarks to the Author):

Recent evidence strongly supports an important role of trans-synaptic nano-columns in synaptic specification. Leucine-Rich Repeat Transmembrane protein LRRTM2, which is a major postsynaptic cell adhesion molecule at excitatory synapses, has been proposed to tether presynaptic molecular release machinery and AMPAR postsynaptic nanodomains via the interaction with presynaptic adhesion molecule Neurexin. Using advanced imaging techniques and LRRTM2 cKO-mediated replacement of endogenous LRRTM2 with their mutants, this study by Liouta et al revealed that the intracellular domain was necessary for the synaptic stabilization of LRRTM2 and AMPARs, while the extracellular domain regulated the presynaptic release machinery organization, AMPAR subsynaptic position, and AMPAR turnover via the Neurexin-binding interface at mature excitatory synapses in cultured hippocampal neurons. The current manuscript is well written, and contains some interesting results. However, the conclusion sounds like a follow-up study to validate the previous cellular or structural biological knowledge on LRRTM2. More detailed mechanisms underlying LRRTM2-mediated stabilization of synaptic AMPARs via its direct or indirect interactions with AMPARs should improve the manuscript significantly. I also found the following points to be improved.

We thank the reviewer for reviewing our paper and for positive appreciation of our work. We show here that the extracellular domain of LRRTM2 drives presynaptic RIM nano-organization and AMPAR synaptic organization and membrane stability through the Neurexin-binding interface containing the residue E348, which has never been studied previously in synapse formation or function. These are important results that shed light on the mechanisms of synapse formation and organization through trans-synaptic adhesion molecules. To answer the reviewer's comment about "more detailed mechanisms underlying LRRTM2-mediated stabilization of AMPARs via direct or indirect interactions", we have tried, like many laboratories to co-immunoprecipitate LRRTM2 and AMPARs in neurons without success, although we are able to detect AMPARs by co-IP in heterologous cells as previously demonstrated (de Wit 2009). To overcome this issue, using our unique cKO model with replacement of endogenous LRRTM2 by AP-tagged LRRTM2, and precipitating exclusively the surface pool of LRRTM2 in neuronal cultures, we were able to detect GluA2 in LRRTM2 surface proteome. We have added these supplementary proteomics data showing that GluA2 subunit of AMPARs is present in the surface proteome of LRRTM2, along with all Neurexin isoforms (1-3) and PSD-95 which are the only known partners identified for LRRTM2 so far. We have added these data as a supplementary figure 6, and discussed them in the main text. Changes are highlighted in blue in the revised manuscript version.

#1 Some imaging data are not convincing. In Figure 1e, YACA, but not dECEV, mutation seems to decrease the number of AP-LRRTM2 and PSD95, compared with the wild type.

- The panel in figure 1e has been changed to show more representative data corresponding to the quantifications, according to the reviewer's suggestion.

#2 The authors should provide scattered plots for individual data points in Figures 2d, 2h, 6d, and 6e.

- We have provided scattered dot plots for these graphs in figures 2 and 6. When the number of points was higher than a few hundreds, we have displayed the data as violin plots to show the distributions.

#3 The current result is based on the replacement of endogenous LRRTM2 with exogenous one, which can potentially cause higher expression levels of LRRTM2 than its endogenous levels. The authors should present a comparison of the expression levels between endogenous and exogenous LRRTM2.

- We have added a supplementary figure showing that the replacement of endogenous by exogenous LRRTM2 does not significantly change the expression levels of LRRTM2 in our assays. This was assessed using

immunolabeling of LRRTM2 and Glyoxal fixation according to Richter et al. 2017. We added a paragraph in the material and methods section (p. 13) and show this data in supplementary figure 2.

#4 The authors should demonstrate the criteria to choose dendritic fragments. The morphology of dendritic branches presented in the current figures seems to differ branch by branch.

- The dendritic fragments were chosen on primary or secondary dendritic branches, depending on experiments. However, neurons in culture have highly variable morphologies, and our experiments rely on transfected cells. We have clarified this point in the material and methods section (p. 14).

#5 The amplitude of mEPSCs in LRRTM2 cKO hippocampal neurons will be helpful to support the authors' imaging result of synaptic GluA expressions, and also compare the synaptic phenotype with previous studies.

- We added these data in supplementary figure 1I, and discussed the discrepancies between studies regarding the roles of LRRTMs in AMPAR-mediated transmission. **We observed a reduction in mEPSC frequency but no change in mEPSC amplitude upon conditional knock-out of LRRTM2 in hippocampal cultures, consistent with previous studies using shRNA knock-down of LRRTM2 (de wit 2009) and double knock-out of LRRTM1 and LRRTM2 in CA1 hippocampal neurons (Dhume 2022).** In the rescue condition however, we observed a slight decrease in mEPSC amplitude, which might be due to partial overexpression of the protein in the recorded cells. However, when we quantified surface protein levels using immunocytochemistry on a large number of cells, we found no significant increase in LRRTM2 expression levels (supplementary figure 2). We added this observation in the results section (p.3).

#6 How did the authors calculate the proportion of the intracellular pools of LRRTM2 at dendritic spines and shafts? Although this must depend on the area they analyzed, no explanation was provided.

- The procedure is now described in the material and methods section entitled “pH change protocol” as followed: “Intracellular pools were extracted as the mean fluorescence under NH₄Cl (total pool) minus the mean of the 3 baseline signals (surface pool). Objects detected at Homer1c-positive spines were considered in spines, where-as the rest were classified as dendritic (shaft).” We have added the following sentence to clarify the calculation: “The proportion of intracellular pools at spines was calculated as the ratio between intracellular pools inside Homer1c-positive spines divided by the total intracellular fluorescence.” (material and methods, p. 15)

#7 What is the molecular mechanism underlying the difference in presynaptic nano-scale organizations of RIM1/2 clusters between EQ and DAT/AA mutations?

- To answer the reviewer's question, **we performed additional experiments** to explore how the mutants affect synaptogenesis and synapse density in mature neurons. First, we had shown that the EQ mutation completely abolishes the binding of purified Neurexin1- β to LRRTM2, unlike the DT/AA mutation which retains Nrxn binding (figure 5c, e). Considering that LRRTM2 induces synapse formation in co-culture assays, we expressed these mutant in COS-7 cells, and examined their potency to induce presynapse formation in wildtype contacting axons. EQ mutation, as well as deletion of the entire extracellular LRR domain prevented presynapse formation in contacting axons compared to WT-LRRTM2. In contrast, DT/AA mutation did not prevent presynapse formation in contacting axons. **These data demonstrate that EQ, but not DT/AA mutation affects presynapse formation and synapsin recruitment in co-culture assays** (data are now presented in revised figure 5 panels f and g). Furthermore, **we quantified synapse density in neurons** expressing the different mutants, and found that only EQ, but not DT/AA mutation leads to reduced synapse density in hippocampal neurons. We have added these data in the revised figure 5 (Figure 5 panels h and i). Finally, **we further quantified RIM sub-synaptic domains (SSDs) per synapse in STORM experiments and show that only EQ – but not DT/AA mutant – disrupts RIM nano-**

organization. These data are now presented in Figure 6, panel e. Altogether, these results show that E348 residue is responsible for the interaction with Nrns and that this interaction in turn is necessary for synapse formation, and the recruitment and organization of the presynaptic machinery. How Nrns are directly linked to RIM nano-organization is still unknown to our knowledge, but a recent paper showed that liprin- α , which interacts with RIM through its coiled-coil regions, clusters Nrns through CASK, a critical step leading to presynapse recruitment and assembly (Marco de la Cruz, Nat. Neuro 2024). **Therefore, in the EQ condition where LRRTM2 fails to bind Nrns, the synaptogenic effect of LRRTM2 is lost, synapse density is reduced, and presynapse nanoorganization is altered. These results show that LRRTM2 trans-synaptically controls presynapse assembly via binding Nrns through E348.** We have added these observations in the results section of the revised manuscript (P. 6-7) and in the discussion section (p. 8-9), as well as a summary model recapitulating our findings now presented in Figure 7.

#8 It is unclear how the authors identify synaptic areas containing both RIM1/2 and AMPAR clusters in dSTORM imaging.

- To identify synaptic areas containing both RIM1/2 and AMPAR clusters in dSTORM experiments, we relied on the low-resolution images of both markers taken before the dSTORM acquisition. We overlaid the low resolution images and chose regions where these signals overlapped. A synapse mask was then generated from these overlapping regions, and RIM1/2 and AMPAR super-resolved domains were analyzed within these regions. We have clarified this point in the material and methods dSTORM section (p. 15).

#9 The authors should explain which they immunolabeled cell-surface or total populations of GluAs in dSTORM imaging in the result section.

- We immunolabeled GluA1/2 surface pools. We have clarified this point in the results section (p. 7).

- Reviewer #3 (Remarks to the Author):

Liouta et al used a conditional knockout mouse model of LRRTM2 and a variety of imaging approaches to assess the role of LRRTM2 in glutamatergic synaptogenesis and receptor stabilization at synapse. The authors first characterized the role of LRRTM2 in synapse development using cKO as well as the mechanism underlying synaptic anchoring and trafficking dynamics of the molecule itself. They then examined the impacts of two mutation sites in LRRTM2 on its interaction with presynaptic Neurexins. Finally they characterized how these mutations modulate the nanoscale organization of synaptic proteins including postsynaptic AMPARs and presynaptic RIM. The experiments in the paper are technically sound and clearly presented, and provide some very interesting and important phenomenon along each lines mentioned above. However, I find many of these findings are incremental, and the whole study is premature and short at a central conceptual advance.

We thank the reviewer for evaluating our manuscript and for positive appreciation of our work. **We have answered the reviewer's comments by performing additional experiments to strengthen our observations.** These are detailed below and in the revised version of our manuscript. We provide here the first demonstration that the critical residue E348 involved in the interaction of LRRTM2 with Neurexin, recently crystallized by Yamagata et al., controls presynapse organization and critically regulates synaptic AMPAR stabilization and nano-organization. The role of this interface has never been addressed in synapse function and organization. Our results show that LRRTM2 trans-synaptically controls presynapse assembly via binding neurexins through E348.

1. The team's previous work as well as others (e.g. Linhoff 2009, Minatohara, 2015) have demonstrated that the YxxC motif at the C-terminal domain of LRRTM2 is critical for its binding with PSD95, synapse positioning and surface diffusion. The data in figures 1-3 in the current study have only verified this in a cKO system in a technically elegant way. The most novel observation in this part is that the YxxC motif regulates the exocytosis of LRRTM2-containing vesicles but the authors did not pursue further on this.

- It appears there may have been a confusion here concerning the different motifs regulating LRRTM2 membrane expression. Indeed, it seems that the YxxC motif was mistaken for the non-canonical PDZ-binding motif ECEV, which is responsible for PSD-95 binding (Linhoff 2009, de wit 2009). The YxxC motif was described in Minatohara 2015, and shown to be involved in controlling LRRTM2 surface expression and clustering. Our previous work (Liouta 2021) showed that the C-terminal region of LRRTM2 is responsible for the protein compartmentalization, addressing it to dendrites and not axons, and that surprisingly, surface diffusion is not dependent on the PDZ-binding motif. Only figure 2 of the present paper recapitulates previous findings, a confirmation we believe necessary considering shRNA potential off-target effects, and difficulty of achieving > 80% knock-down efficacy. Also, at a time when replicability in science is a topic of great concern, replicating previous findings with the best available methodology seems important. In contrast, **figures 1 and 3 both display novel results that have not been published previously.** Figure 1 shows that LRRTM2 is clustered at synapses through intracellular interactions, independent from the extracellular LRR domain, and figure 3 explores for the first time the role of the YxxC motif in LRRTM2 exocytosis, a topic never addressed previously. The reviewer states that figure 3 is a simple verification of previous findings, and then that "the most novel observation in this part is that the YxxC motif regulates the exocytosis of LRRTM2-containing vesicles", which is precisely the data presented in figure 3. The investigation of LRRTM2 exocytosis and membrane trafficking are being pursued and are out of the scope of the present study. Here, **our most novel and important finding is that LRRTM2 trans-synaptically controls synapse organization through the Nrnx-binding site E348 (discovered in 2018 by Yamagata et al.). The role of this interaction interface in synapse formation and function has never been addressed previously.**

2. The role of LRRTM2 in stabilizing AMPARs at synapses is well established by multiple studies, e.g. Soler-Llavina 2013, Aoto 2013, Bhouri 2018, and Ramsey 2021. The FRAP results in figures 4 and S5 are well predicted.

Moreover, considering the CTD is key for synaptic targeting of LRRTM2, the data from neurons expressing LRRTM2 without CTD add no meaningful information on the modulation of AMPAR stabilization.

- The role of LRRTM2 in stabilizing AMPAR at synapses is indeed well established by previous studies cited here by the reviewer. However, no one has examined the role of LRRTM2-neurexin binding interface on AMPAR stabilization and presynapse organization, because the binding interface between LRRTM2 and neurexin seems to have been wrongly assumed (Siddiqui 2011 and Paatero 2016, but see Yamagata 2018). The complex between **LRRTM2 and Neurexin1- β was crystallized in 2018** by Yamagata et al, showing that the E348 residue is critical for this interaction and that the previously identified residues D259/T261 are not involved in this interaction. **The papers cited here by the reviewer are all anterior to this date (except Ramsey 2021, which has not addressed this point), and have all relied on the wrongly assumed binding residues D259/T261. Therefore, our study is the first one to explore the importance of LRRTM2-Neurexin binding interface in synapse formation and function,** and shows that LRRTM2 controls synapse organization and AMPAR stabilization through this interface. Regarding the contribution of the CTD to AMPAR stabilization, it is well known that synaptic AMPAR stabilization relies on PSD-95 interactions through auxiliary proteins (Bats, Neuron 2007, Chen PNAS 2015), and we show here that LRRTM2 cKO or CTD deletion reduces PSD-95 density. Thus, in the absence of LRRTM2, it is not clear whether the destabilization of AMPARs relies on the missing PSD-95 slots or on the direct stabilization by LRRTM2 extracellular domain. We have addressed this issue by comparing the effects of intra- and extra- cellular domain deletion on the stabilization of synaptic AMPARs in figure S5 (figure S7 in the revised version). Our results show that CTD deletion does not significantly affect AMPAR slow pool fraction in FRAP experiments (we have added these data in panel e of supplementary figure 7 to emphasize this point), but deletion of the LRR domain strongly affects AMPAR stabilization (Fig. S7e). **These results demonstrate that it is the extracellular domain of LRRTM2 that stabilizes AMPARs at synapses, and not an indirect effect through the recruitment of PSD-95 scaffolds.**

3. For the FRAP results, the recovery time constant and extent should be quantified and discussed. Also, with the small differences between groups, another control experiment in neurons with intact endogenous LRRTM2 should be performed.

- Following the reviewer's suggestion, we fitted and quantified FRAP data according to Makino and Malinow 2009, and added the corresponding data and graphs with statistical analyses in each figure (Fig. 3c, d; Fig. 4c, d, g, h; Fig. 5m, n; Fig. S4e, f; Fig. S7d-f; Fig. S8c, d). We added a section in the material and methods to describe these analyses (p. 15) and discussed the results in the main text (result section, p. 5, p.6). Regarding the reviewer's suggestion to perform control FRAP experiment with intact endogenous LRRTM2: if the reviewer is referring to FRAP of SEP-GluA1 or SEP-GluA2, the "control" conditions in figure 4 precisely correspond to control FRAP experiments with intact endogenous LRRTM2. For the other experiments where LRRTM2 itself is frapped, it is not possible to our knowledge to perform FRAP on intact endogenous LRRTM2 because FRAP experiments rely on the use of fluorescence, which is usually absent from endogenous proteins.

4. To characterize the effects of the two mutants on Nr_x binding, a simple binding assay with purified Nr_x fragment is not convincing. A synaptogenesis test with cocultures and a quantification of synapse density in mutant-expressing neurons would provide much more information.

- We thank the reviewer for this insightful comment and for the suggestions to improve and strengthen our observations. We have performed additional experiments according to his/her remarks. In complement to the widely used Nr_x-binding assay, **we performed synaptogenic test using co-cultures of COS-7 cells and hippocampal neurons and quantified synapse density in neurons** expressing the different mutants, as suggested. Our synaptogenic assays show that deletion of the LRR domain or mutation of E348 both prevent synapse formation in co-culture assays to the same extent, whereas DT/AA mutation does not significantly alter synapse induction compared to WT-LRRTM2. We have now added these data in **revised figure 5**, panels f and g, presented

them in the result section p. 6, and discussed these in the discussion section p. 8. Since this synaptogenic effect relies on Nrnx-binding, this is further evidence that DT/AA mutant can still bind Nrnxns and induce synapse formation, contrary to EQ mutant which disrupts Nrnx-binding. Following the reviewer's suggestion, we further quantified synapse density in mutant-expressing neurons using immunolabeling of pre- and post- synaptic markers and found that EQ mutation significantly reduced synapse density, compared to WT- and DT/AA- LRRTM2. We have added these experiments in revised figure 5h, i; and discussed them in the main text (p. 6).

5. LRRTM2 has been proposed to stabilize AMPARs through direct binding and/or indirect scaffolds-mediated interactions. The authors got around this key question and measured how the mutations related to Nrnx binding affects AMPAR stabilization. However, without a solid conclusion on how the mutants impact the LRRTM2-Nrnx interaction and the synapse formation, these results lead to no deeper understanding of the mechanism underlying receptor dynamics.

- **We now have a solid conclusion on how the mutants affect LRRTM2-Nrnx interaction (see point 4)**, thanks to the reviewer's suggestions addressed in the previous points by adding experimental data to strengthen our observations. Additionally, we have partially addressed this question already in point 2, following the reviewer's statement that "the data from neurons expressing LRRTM2 without CTD add no meaningful information on the modulation of AMPAR stabilization". **The present point 5 was directly addressed in Figure S5 of the original manuscript (now Figure S7 in the revised manuscript)**, but we might not have sufficiently explained the rationale for this figure. Our results show that AMPAR-dependent stabilization does not rely on indirect scaffold-mediated interactions, but rather on extracellular interactions involving the LRR domain (figure S7c-e, and see point 2 of the present answer). Thus, LRRTM2 stabilizes AMPARs through direct or indirect binding occurring in the extracellular domain of LRRTM2. Binding through the LRR domain has been detected in heterologous cells (de Wit 2009, Soler-Llavina 2013), but never in neuronal preparations, although many laboratories, including our own, have tried. **Using a unique strategy to target LRRTM2 surface proteome in our cKO model, we were able to detect GluA2 subunit in LRRTM2 surface proteome, along with all Neurexins isoforms, and PSD-95, which are the only known interactors** of LRRTM2 so far. These results may indicate direct or indirect interaction, but remain to be further investigated. **We have added these data in supplementary figure 6** of the present manuscript, result section (p. 5), discussion section (p.9) and added a paragraph in the material and method section (p. 12).

6. The authors quantified many detailed changes in nanoscale organizations in figure6. What do these structural changes mean for synaptic functions?

- To address the functional effects of preventing Neurexin binding to LRRTM2 by specifically mutating the critical residue E348 involved in this interaction, **we performed electrophysiological recordings**. Hippocampal neurons were transfected with the different mutants and miniature Excitatory Post-Synaptic Currents (mEPSCs) were recorded. **Our results show that Neurexin-deficient mutant significantly affects the amplitude distributions of recorded mEPSCs (new supplementary Fig. S9)** and only DT/AA affects the frequency distribution of mEPSCs. No effects were observed on average mEPSC amplitudes or frequencies, suggesting a non-uniform effect on individual events that is not detected when averaging the whole population. Considering all the data together, it seems that the smallest mEPSC events (< 15 pA) are lost in the mutant conditions, therefore affecting the amplitude distributions towards larger values. We have added a supplementary figure (Fig. S9) with these data and discussed the results in the main text (results section p. 7; discussion section, p.9).

7. What are the functional impacts of these LRRTM2 mutants?

- We have already addressed this question in points 6 and 4 of the present answer, by performing additional electrophysiology experiments which are now presented in supplementary figure 9, and additional synaptogenesis tests and quantification of synapse density in the mutant conditions now presented in figure 5

(panels f, g, h, i). **To sum up, the functional impact of disrupting Neurexin binding to LRRTM2 by mutating E348, is a failure to induce the formation of synapses and a loss of presynapse organization, as well as a destabilization of AMPARs and their nanoscale organization, leading to impairments in mEPSC amplitude distributions.**

8. what is the expression level of the AP-tagged LRRTM2 variations compared with the endogenous protein?

- We have added a supplementary figure showing that the replacement of endogenous by exogenous LRRTM2 **does not significantly change the expression levels** of LRRTM2 in our assays (supplementary figure 2).

9. The mixture use of different abbreviations for the same mutation is disturbing, e.g. EQ vs E348Q, DT/AA vs D259/T261, YACA vs Y500A/C503A.

- We have addressed this issue by homogenizing the occurrences. The mutants are now "EQ", "DT/AA" and "YACA", and we used the numbers corresponding to the amino acids (E348, D259/T261, Y500/C503) only to indicate the positions of the mutations.

REVIEWER COMMENTS

Reviewer #1 (Remarks to the Author):

The manuscript, both text and figures, has been extensively revised, I'm happy with the changes, and I have no further concerns. This is, in my opinion, a very strong manuscript fully deserving of publication.

Reviewer #2 (Remarks to the Author):

I appreciate the authors' efforts regarding my concerns. I am satisfied with the majority of the authors' responses. However, I still find the conceptual advance of the revised manuscript somewhat insufficient. Figure 6 and supplementary Figures 7-9 clearly indicate a mechanism for controlling AMPAR stability that does not involve the intracellular domain of LRRTM2. To support this, the authors conducted a new proteome analysis to identify interactors with surface LRRTM2. However, this analysis lacks validation, and it would be beneficial at least to share the list of proteins identified from the analysis. Addressing this mechanism would be particularly intriguing.

Reviewer #3 (Remarks to the Author):

I appreciate the authors' efforts in conducting additional experiments and analyses to address the previous comments. The new results have provided a list of very interesting findings that significantly strengthen the manuscript.

Overall, the paper has characterized the role of major domains and key residues of LRRTM2 in the synaptic targeting of the molecule itself, as well as the molecule-mediated nanoscale organization of synaptic architectures. I would strongly suggest the authors revise the abstract to better represent the key novelties and the broader insights gained from the characterizations. Currently, the abstract sets up the story based on nanocolumn modulation and places too much emphasis on the E348 residue, as claimed in the rebuttal letter that the central novelty is the demonstration of the critical role of the E348 residue at the LRRTM2-Neurexin interface in controlling synaptic nanoorganization. While this is an important finding, I don't believe it alone brings enough conceptual novelty for publishing in a high-profile journal like Nature Communications, as this could be reasonably predicted based on previous findings that: 1) the LRRTM2 extracellular domain is essential for AMPAR nanoorganizations and release-AMPA alignment, 2) Neurexin is the only known presynaptic binding partner of LRRTM2, and 3) the E348 residue of LRRTM2 is key for interacting with Neurexin.

Regarding the statistical analysis, please provide more details on how the fractions were compared in Figures 2f and 2i.

The calculated slow pool fraction in the Cre+WT group in Figure 4h and the DT/AA group in Figure 5m-n appear quite unexpected based on the trends shown in the corresponding Figures 4g and 5m-n. Please provide a more detailed explanation.

Finally, I suggest moving the electrophysiology data from Figure S9a,b to the main figure panels to better highlight the importance of these findings.

Point-by-point answer to the reviewer's comments

Reviewer #1 (Remarks to the Author):

The manuscript, both text and figures, has been extensively revised, I'm happy with the changes, and I have no further concerns. This is, in my opinion, a very strong manuscript fully deserving of publication.

We thank the reviewer for their positive feedback on the manuscript and extensive revisions, for reviewing the present paper and supporting our work.

Reviewer #2 (Remarks to the Author):

I appreciate the authors' efforts regarding my concerns. I am satisfied with the majority of the authors' responses. However, I still find the conceptual advance of the revised manuscript somewhat insufficient. Figure 6 and supplementary Figures 7-9 clearly indicate a mechanism for controlling AMPAR stability that does not involve the intracellular domain of LRRTM2. To support this, the authors conducted a new proteome analysis to identify interactors with surface LRRTM2. However, this analysis lacks validation, and it would be beneficial at least to share the list of proteins identified from the analysis. Addressing this mechanism would be particularly intriguing.

We thank the reviewer for reviewing our manuscript and for their positive feedback. We are pleased that the revisions have met their approval. Regarding the new proteome analysis, peptide validation was performed using Percolator (Käll et al., Nat. Methods 2007) and only high confidence peptides were retained corresponding to 1% False Positive Rate at peptide level (indicated in the material and methods section). The list of proteins represented in the supplementary figure is now displayed as supplementary table 1. To further cross-validate these data, and investigate the interaction between LRRTM2 and AMPARs, we performed LRRTM2 streptavidin pull-down assays in COS-7 cells expressing AP-LRRTM2 and SEP-GluA2. WT-LRRTM2, but not Δ LRR-LRRTM2, co-precipitated GluA2. We added these data in Fig. S6c. In parallel, we performed co-recruitment assays in living cells to determine whether GluA2 and LRRTM2 co-segregate in a cellular context. We cross-linked SEP-GluA2 receptors at the surface of COS-7 cells and examined the co-aggregation of LRRTM2. WT-LRRTM2 efficiently co-segregated with cross-linked GluA2 in these experiments, as seen by the formation of colocalized clusters. In contrast, Δ LRR-LRRTM2 failed to co-segregate with GluA2. We added these data in Fig. S6d. Changes appear in blue in the main text.

Altogether, these experiments clearly indicate that GluA2 and LRRTM2 physically associate and are part of the same molecular complexes in neurons as well as in heterologous cells, and validate our proteomics data.

Finally, we have uploaded the original proteomics data on the ProteomeXchange Consortium via the PRIDE partner repository with the dataset identifier PXD054623. We have indicated this in the data availability section in the material and methods page 17. Changes appear in blue.

Reviewer #3 (Remarks to the Author):

I appreciate the authors' efforts in conducting additional experiments and analyses to address the previous comments. The new results have provided a list of very interesting findings that significantly strengthen the manuscript.

We thank the reviewer for reviewing our manuscript, and appreciate their positive feedback on the manuscript and extensive revisions which have significantly strengthened our study.

Overall, the paper has characterized the role of major domains and key residues of LRRTM2 in the synaptic targeting of the molecule itself, as well as the molecule-mediated nanoscale organization of synaptic architectures. I would strongly suggest the authors revise the abstract to better represent the key novelties and the broader insights gained from the characterizations. Currently, the abstract sets up the story based on nanocolumn modulation and places too much emphasis on the E348 residue, as claimed in the rebuttal letter that the central novelty is the demonstration of the critical role of the E348 residue at the LRRTM2-Neurexin interface in controlling synaptic nanoorganization. While this is an important finding, I don't believe it alone brings enough conceptual novelty for publishing in a high-profile journal like Nature Communications, as this could be reasonably predicted based on previous findings that: 1) the LRRTM2 extracellular domain is essential for AMPAR nanoorganizations and release-AMPA alignment, 2) Neurexin is the only known presynaptic binding partner of LRRTM2, and 3) the E348 residue of LRRTM2 is key for interacting with Neurexin.

We thank the reviewer for highlighting that the abstract should be revised given our experimental additions to the manuscript. We have followed their suggestion to better present the key findings and novelties of our study. The abstract has been modified accordingly. The changes appear in blue.

Regarding the statistical analysis, please provide more details on how the fractions were compared in Figures 2f and 2i.

To compare the fractions, we first extracted their values for each cell, based on the mean square displacement of single molecules described in the material and methods. We then performed a two-way analysis of variance test to compare the fractions of immobile, confined or diffusive tracks between the different conditions. We have added a sentence in the statistics part of the material and methods page 17 to specify this point. Changes appear in blue.

The calculated slow pool fraction in the Cre+WT group in Figure 4h and the DT/AA group in Figure 5m-n appear quite unexpected based on the trends shown in the corresponding Figures 4g and 5m-n. Please provide a more detailed explanation.

In these datasets, each FRAP curve corresponding to AMPAR fluorescence recovery in bleached regions was fitted as previously described for AMPARs by Makino and Malinow (Neuron 2009). The procedure and equation are described in the material and methods "FRAP" section, pages 15-16. This equation describes a two-state model with a fast recovering pool accounted for by fast diffusion into the bleached area, and a slower recovery pool due to slower molecule dynamics that can be influenced by factors such as molecular binding or restricted diffusion. For each recovery curve, we extracted a slow pool fraction from the fits performed in GraphPad Prism using the equation described in the material and methods. This

corresponds to the slow recovering pool of AMPARs. The distributions of slow pool fractions per condition can be seen on the graphs, as individual points are plotted in figures 4h and 5n. Statistical significance was tested by one-way analysis of variance test, followed by post hoc Dunn's test, as described in the statistics paragraph. We have added a sentence in the FRAP section of the material and methods to clarify this point, page 16. Changes appear in blue.

Finally, I suggest moving the electrophysiology data from Figure S9a,b to the main figure panels to better highlight the importance of these findings.

We thank the reviewer for this observation and have followed their suggestion. The electrophysiology data are now displayed in main figure 6.

REVIEWERS' COMMENTS

Reviewer #2 (Remarks to the Author):

I appreciate the authors' efforts regarding my concerns again. The revised manuscript has fully satisfied me, and I have no further issues. I hope it provides readers with new insights into the molecular organization of LRRTM2-mediated transsynaptic nanocolumns.

Reviewer #3 (Remarks to the Author):

All major concerns of mine have been address.

Minor:

Line 26 page 1, "Surface expression, synaptic clustering, and membrane dynamics of AMPARs are..."

Point-by-point response to reviewer's comments

Reviewer #2 (Remarks to the Author):

I appreciate the authors' efforts regarding my concerns again. The revised manuscript has fully satisfied me, and I have no further issues. I hope it provides readers with new insights into the molecular organization of LRRTM2-mediated transsynaptic nanocolumns.

- We thank the reviewer for their appreciation and evaluation of our work.

Reviewer #3 (Remarks to the Author):

All major concerns of mine have been address.

Minor:

Line 26 page 1, "Surface expression, synaptic clustering, and membrane dynamics of AMPARs are..."

- We thank the reviewer for their appreciation and evaluation of our work.